# Causal3D: A Comprehensive Benchmark for Causal Learning from Visual Data

## Abstract

True intelligence hinges on the ability to uncover and leverage hidden causal relations. Despite significant progress in AI and computer vision (CV), there remains a lack of benchmarks for assessing models' abilities to infer latent causality from complex visual data. In this paper, we introduce **Causal3D**, a novel and comprehensive benchmark that integrates structured data (tables) with corresponding visual representations (images) to evaluate causal reasoning. Designed within a systematic framework, Causal3D comprises 19 3D-scene datasets capturing diverse causal relations, views, and backgrounds, enabling evaluations across scenes of varying complexity. We assess multiple state-of-the-art methods, including classical causal discovery, causal representation learning, and large/vision-language models (LLMs/VLMs). Our experiments show that as causal structures grow more complex without prior knowledge, performance declines significantly, highlighting the challenges even advanced methods face in complex causal scenarios. Causal3D serves as a vital resource for advancing causal reasoning in CV and fostering trustworthy AI in critical domains.

## 1 Introduction

Computer vision (CV) has achieved remarkable success in tasks such as classification (Dosovitskiy et al., 2021; Singh et al., 2022; Fang et al., 2023) and detection (Liu et al., 2021; Wang et al., 2023). Although these systems excel at identifying statistical correlations within data, they often struggle to infer deeper causal relations. This limitation significantly impacts their ability to be applied to high-stakes domains or unseen scenes. For instance, without understanding the causal relations between object depths, motions, and shapes, a vision-based autonomous driving system may easily misidentify traffic signs due to spurious correlations or adversarial attacks, leading to sudden braking and severe safety issues (Yang et al., 2022).

Unlike classification and detection tasks, which have thrived on large-scale datasets with explicit labels, causal tasks in images demand more for study and evaluation — ideally, annotations with clear causal relations among variables. This makes dataset collection significantly more challenging than in traditional CV tasks. The difficulty stems from two key factors: **inherent complexity and covert nature of causality**: Real-world causal relations are often complex and not directly observable, and causal variables are often high-level concepts (e.g., an object) instead of low-level pixels, making causal relations in vision domain inherently challenging to discern; **challenges in visual representation**: Even well-established causal rules are challenging to visually depict. For example, in physics, magnetic fields are represented by invisible magnetic induction lines, making them difficult to illustrate in realistic images. Similarly, abstract concepts like economic principles, (e.g., supply and demand), are not easily encoded into visual forms.

Some existing datasets have been involved in causal studies in visual AI systems. However, these datasets often have significant limitations. For instance, oversimplified 2D hypothetical datasets (Yang et al., 2020) fail to capture the richness and complexity of real-world environments. Similarly, domain-specific datasets like the CelebA face dataset (Liu et al., 2015) are not designed for causal reasoning and lack the structural diversity required for comprehensive explorations. Recent datasets developed for vision- and multimodal-language models (VLMs/MLMs) (Zimmermann et al., 2021; Von Kügelgen et al., 2021; Mao et al., 2022; Tung et al., 2024) have improved in complexity and realism but remain limited in explicitly representing causality and in offering diverse

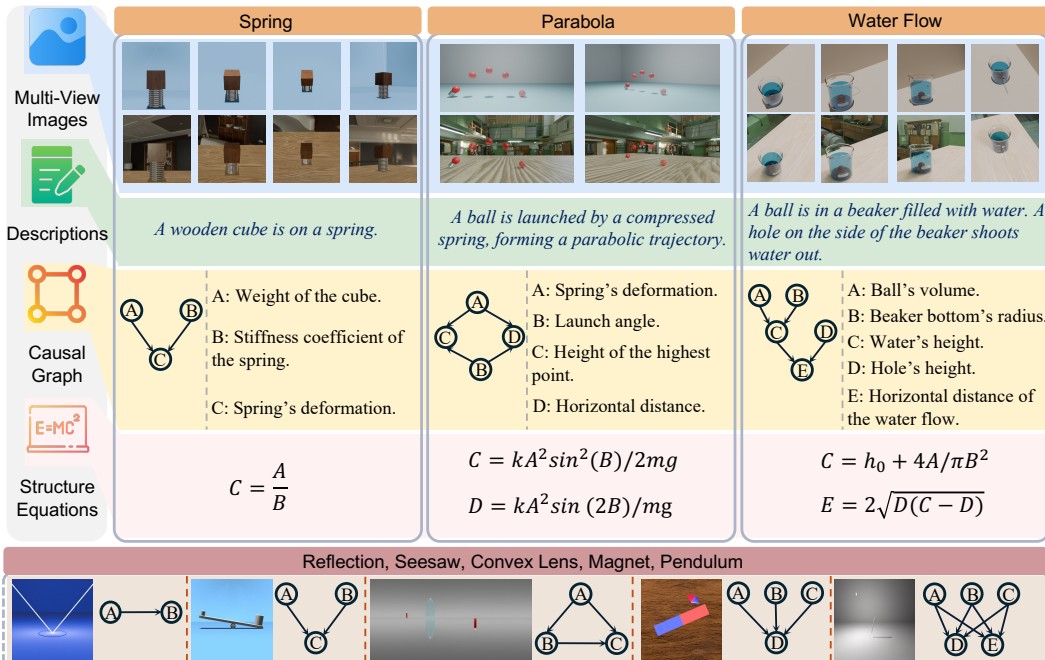

Figure 1: The proposed CAUSAL3D dataset. We display 8 real-world scenes (11 hypothetical scenes are in the Appendix 6). We focus on 3 scenes: springs, parabolas, and water flow. 1) The blue block represents multi-view images of each scene, offering four different views. The first row shows virtual backgrounds, while the second row shows real backgrounds, with the same view in each column; 2) The green block provides textual descriptions; 3) The yellow block represents the causal graphs for each scene, along with the meanings of each variable in the graphs; 4) The pink block shows the structural equations (i.e., functions describing causal relations) for each scene. The bottom row briefly presents an overview of the remaining 5 real-world scenes and their corresponding causal graphs, including reflection, seesaw, convex lens, magnet, and pendulum. Detailed information on these 5 scenes can be found in the Appendix 6.

causal relations. The absence of clearly defined causal relations within visual representations, along with the lack of tabular records that are tightly aligned with these representations to provide guidance, makes such physics-aware VLMs/MLMs datasets suboptimal for fine-grained causal reasoning tasks. Especially, as interest in 3D data grows, causal learning in 3D settings introduces new challenges, opportunities, and insights. The complexity of realistic 3D scenes—encompassing lighting, texture, background, and viewpoint—can introduce spurious correlations and backdoor paths, making causal inference more difficult. At the same time, 3D environments provide multi-view consistency, allowing models to observe the same underlying causal relationships from diverse perspectives, which helps disentangle true causality from viewpoint-dependent features. This makes **3D datasets uniquely valuable for developing and evaluating robust, causally grounded models in realistic settings**. However, this area remains underexplored.

In general, existing visual datasets either lack explicit definitions of causal relations beyond visual representation or are too specific and simplified to enable comprehensive exploration of diverse causal relations. On the other hand, datasets in the causal research community, while rich in diverse causalities and clear causal definitions, lack corresponding visual representations, making them unsuitable for tasks involving causal reasoning in images, not to mention complicated 3D scenarios. This disconnect makes it challenging to effectively advance and evaluate AI systems' abilities of reliable reasoning, thereby creating a significant bottleneck in advancing this field.

To address these limitations, we introduce **CAUSAL3D**, the first benchmark specifically designed to systematically explore and evaluate causality learning through a combination of realistic 3D imagery and explicit causal structures (*i.e.,* causal graphs and structure equations). Using CAUSAL3D, we conduct experiments on existing algorithms and tools, **establishing a comprehensive benchmark** to evaluate models' abilities to identify and leverage diverse causal relations.

Table 1: Qualitative comparison between Causal3D and other causality related dataset. "Diverse Structure" refers to whether the dataset covers different causal graph structures, e.g., our dataset involves 13 different graph structures spanning from real and hypothetical scenes.

| Dataset | Dual Representation of Causality | | Explicit Causal Structures | | Hybrid Causal Framework | | Diverse Structure and 3D Scenes | |
|---|---|---|---|---|---|---|---|---|
| | Tabular | Visual | Linear | Nonlinear | Physical Consistent | Hypothetical Scenes | Diverse Structure | Diverse 3D Scenes |
| CauseMe (Runge et al., 2019a) | ✓ | ✗ | ✓ | ✓ | ✗ | ✓ | ✗ | ✗ |
| CelebA (Liu et al., 2015) | ✓ | ✓ | ✗ | ✗ | ✗ | ✓ | ✗ | ✗ |
| CausalVAE (Yang et al., 2020) | ✓ | ✗ | ✓ | ✓ | ✓ | ✓ | ✗ | ✗ |
| Causal3DIdent (Zimmermann et al., 2021) | ✗ | ✓ | ✓ | ✗ | ✓ | ✗ | ✗ | ✓ |
| Craft (Ates et al., 2022) | ✗ | ✗ | ✓ | ✗ | ✓ | ✗ | ✗ | ✗ |
| CLEVR-Humans (Mao et al., 2022) | ✗ | ✓ | ✓ | ✗ | ✓ | ✗ | ✗ | ✓ |
| Physion++ (Tung et al., 2024) | ✗ | ✓ | ✓ | ✓ | ✓ | ✗ | ✗ | ✓ |
| **Causal3D** | ✓ | ✓ | ✓ | ✓ | ✓ | ✓ | ✓ | ✓ |

To the best of our knowledge, CAUSAL3D (see Fig. 1) is the first dataset tailored for causality studies that combines realistic 3D scenes with explicit causal graphs (i.e., Directed Acyclic Graphs (DAGs) representing variables as nodes and causal relations as edges). This dataset stands out due to several key features: **1) Dual Representation of Causality**: CAUSAL3D provides a dual representation of causality by including both tabular data (for high-level concepts) and strongly corresponding visual representations in multiple 3D scenes. This design provides sufficient information for causal studies in vision, enabling precise evaluations of models in related domains. **2) Diverse Design**: The difficulty in CAUSAL3D is diversely structured, derived from different dimensions, including the number of variables (ranging from 2 to 5), multiple causal structures, different (linear/nonlinear) types of causal relations, and various camera views and backgrounds in 3D scenes. This design enables benchmarking with progressively increasing levels of challenges, allowing for fine-grained evaluation of model performance. **3) Physically Consistent and Hypothetical Scenes**: CAUSAL3D encompasses both real-world and hypothetical scenes. By leveraging established physical rules, it creates datasets with realistic causal relations, enhancing the dataset's authenticity. To further diversify causal scenes, CAUSAL3D incorporates hypothetical causal relations, providing a broader range of possibilities and enriching its utility for causality research. With the dataset, we designed systematic experiments to evaluate representative causal methods and LLMs/VLMs in different causal tasks. In this work, our primary contributions are threefold:

- **Dataset** We introduce CAUSAL3D, a novel and comprehensive benchmark consisting of 19 datasets that span a wide range of causal structures, viewpoints, and background variations within realistic 3D scenes. The full dataset will be released to support further research.

- **Evaluation** We implement a thorough evaluation of models on CAUSAL3D, spanning from traditional causal algorithms to advanced LLMs and VLMs for images, offering a detailed analysis of the current state-of-the-art models on our benchmark.

- **Insights** We lay a strong foundation for advancing causal learning in CV by bridging the gap between these fields through our benchmark and provide key insights from our experimental observations.

## 2 RELATED WORK

**Causal Discovery from Tabular Data** Causal discovery is an important task in causal inference (Pearl, 2009), aiming to identify the causal relations from data. Multiple methods have been developed for this task and most of them focus on tabular data (Wen et al., 2021; Tu et al., 2024; Wen et al., 2022; Cinquini et al., 2021; Russo and Toni, 2023). These methods mainly include constraint-based methods (e.g., PC (Spirtes et al., 2000b)) and score-based methods (e.g., GES (Chickering, 2002)). Many of them are statistical methods (e.g., CAM (Bühlmann et al., 2014), LiNGAM (Yang et al., 2024)), which have good theoretical support but often suffer from strong assumptions, sensitivity, and scalability issues.

Recently, deep learning-based methods have attracted lots of attention due to their improvement in these aspects. Among them, causal-TGAN (Wen et al., 2022), GraN-DAG (Lachapelle et al., 2020), DAG-GNN (Yu et al., 2019)) can model nonlinear causal relations in large datasets. Diffusion-based approaches (e.g., DiffAN (Sanchez et al., 2023)) improve robustness against noises while being computationally intensive and sensitive to hyperparameters.

**LLM-based Causal Discovery** Recent advances in LLMs have broadened their role in causal discovery (Ma, 2024; Jin et al., 2024; Liu et al., 2024b; Ban et al., 2023; Liu et al., 2024a; Wu et al., 2024; Wan et al., 2024; Shen et al., 2024). LLM-based causal discovery spans pairwise and full-graph discovery (Ma, 2024; Kıcıman et al., 2023; Jiralerspong et al., 2024), often leveraging prompts such as binary or multiple-choice selection and natural question-answering. Among them, many state-of-the-art models like ChatGPT 4o (Rawal et al., 2024) and Gemini-1.5 Pro (Carro et al., 2024) have been widely explored for causal inference. Besides, some causality-specific agents like Causal Copilot (Wang et al., 2024b) integrate LLMs for natural language-based causal queries. Despite its promising performance, LLMs still face key limitations, including difficulty in handling latent confounders and complex causal tasks.

**Causal Methods in CV** Causal inference in CV is essential for improving generalization and interpretability (Yang et al., 2021; Schölkopf, 2022). Many causal tasks have thus been widely explored in image data, one of them is causal representation learning, which aims to uncover disentangled and causally meaningful representations corresponding to high-level variables from data (Liu et al., 2022; Deng et al., 2022; Schölkopf et al., 2021). Many representative approaches in this area are based on generative models (e.g., CausalVAE (Yang et al., 2023), DEAR (Shen et al., 2022)). However, they often depend on strong assumptions (e.g., available annotation of high-level concepts) that may not always hold. Recently, explorations of VLMs in causal tasks under more general scenes have also attracted increasing attention (Wang et al., 2024a; Zhao et al., 2024).

**Causal Datasets in CV** Tabular data has long dominated causal inference research (Runge et al., 2019a; Runge, 2018; Runge et al., 2019b; Spirtes et al., 2000a; Zheng et al., 2018). With the recent increasing need for causal studies in different data types and modalities, the community in CV and VLM has also placed more emphasis on causality. Recent years have witnessed the emergence of image and video datasets for causal reasoning (Zimmermann et al., 2021; Ates et al., 2022; Mao et al., 2022; Tung et al., 2024), bridging the gap between CV learning and causal reasoning. Despite these datasets addressing the absence of visual data encoded with causality, most of them still remain limited. They either focus on specific scenes, restricting the diversity of causal relations or lack rigorous causal definitions—such as explicit causal graphs and structured tabular records—to capture interactions among in-image variables. Consequently, a gap remains between traditional causal research and the study of causality in CV and VLM, as summarized in Tab. 1. This highlights the need for datasets that integrate explicit causal structures with both visual and tabular representations.

## 3 CAUSAL3D: THE PROPOSED BENCHMARK

We introduce CAUSAL3D, a realistic 3D image dataset designed for casual learning from *observational* visual data. Aiming to bridge the gap between causal study and CV/VLM community, CAUSAL3D is established with structured tabular data and tightly aligned visual representation. To build a comprehensive benchmark for evaluating models' ability to uncover causality, CAUSAL3D contains visual representations encoded with diverse causal relations among multiple variables. The dataset comprises two main components: **Physically Consistent 3D Scenes**, which simulate real-world settings to enhance the authenticity of the dataset, and **Hypothetical 3D Scenes**, introduced to diversify the causal relations represented in the dataset. To simulate realistic images in 3D scenes, we select or design causal relations to generate structured datasets and use Blender[1] to render high-quality images, as shown in Fig. 2. Furthermore, by introducing various views and backgrounds, CAUSAL3D presents the same scene in different surroundings, enhancing dataset diversity and contextual richness.

### 3.1 DATASET COMPONENTS

**Physically Consistent Scenes** In the physically consistent setting, CAUSAL3D incorporates fundamental physical principles, such as light dynamics, magnetic fields, water pressure, and mechanics, to ensure realistic causal relations. To systematically vary the complexity of causal discovery, we curated 8 distinct scenes, each featuring 2 to 5 variables with unique causal structures. Each scene contains 10K samples. We provide visual overviews of each scene in Appendix 6.

**Hypothetical Scenes** Since causal relations in reality are often complex and difficult to observe, designing scenes with diverse causal graphs is challenging. In CAUSAL3D, we also introduce addi-

---

[1]https://www.blender.org/

tional hypothetical scenes that broaden the range of causal relations in our benchmark. Specifically, we explore causal relations under artificially defined hypothetical rules and synthesize causal relations among three fundamental 3D objects in Blender: sphere, cuboid, and cone. By defining specific dependencies among their variables (*e.g.,* sphere radius, cuboid height), we construct both linear and non-linear causal relations across various graph configurations. This process yields 11 hypothetical scenes, each containing 10K samples. More details can be found in Appendix 6.

## 3.2 DATA CONSTRUCTION

The data construction process is shown in Fig. 2. The upper half shows the construction of real physical scenes. We first collect physical entities that exist in the real world, such as springs and magnets, etc. Then, we explore the physical laws within these entities and identify the corresponding causal graphs. Based on the causal graphs, we can generate tabular data, where each row represents a sample, and different columns correspond to different values of various variables. For variables without parents, we assign values randomly using a uniform distribu-

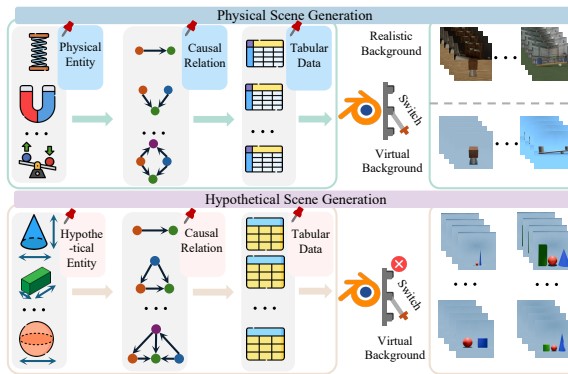

Figure 2: Illustration of data construction pipeline.

tion, while the values of other variables are calculated according to the causal graphs and physical laws. For example, if we select a spring and a block as our physical entities, the variables involved include the spring constant $k$, the deformation $X$, and the weight of the block $W$. The physical law governing these variables is Hooke's Law: $X = W/k$. We can randomly assign values to $W$ and $k$, and then calculate $X$ accordingly. The generated tabular data can then be input into Blender to render multi-perspective scenes. It is worth noting that we have set a background switch to choose whether to use a real or virtual background.

The lower half of Fig. 2 displays the construction of hypothetical scenes. We first identify various dimensions of geometric bodies to serve as variables and then manually design the relations among these variables. The remaining steps are the same as those in generating physical scenes, with the only difference being that they do not include real backgrounds because the hypothetical objects and relations do not exist in the real world.

## 3.3 TASK DESIGNS

Based on our dataset, we focus on three key causal tasks to evaluate state-of-the-art algorithms and models in discovering and leveraging causal relations across diverse causal structures and scenes. We focus on the widely adopted setting of causal learning from observational data. These tasks include:

**Causal discovery from tabular data**  This task focuses on identifying latent causal relations among variables using only tabular data. In this setting, we have high-level causal variable values recorded in the tabular data and do not rely on image information. It is evaluated based on the correctness of inferred causal structures across various datasets and underlying causal mechanisms.

**Causal representation learning from images**  This task aims to learn disentangled and causally meaningful representations from images, meanwhile enabling models to capture underlying causal relations between high-level concepts. We use image data along with any additional information required by the models as input. Evaluation is based on generated images after intervening on learned representations, assessing whether they accurately reflect the corresponding causal variables and causal relations.

**Causal discovery & intervention from few images**  This task focuses on uncovering causal relations and assessing intervention results with a limited number of images. Unlike traditional causal discovery methods that rely on large datasets, this task evaluates the ability of models to infer causal structures from a small set of images. Furthermore, we conduct causal interventions by manipulating a certain variable to observe its effect on the whole image. Causal intervention is often implemented with the do-operator $do(\cdot)$ (Pearl, 2009). Here, $do(X = x)$ means modifying a variable $X$ to a specific value $x$ while keeping all other influences unchanged. Intervention evaluation is based on whether the intervened images still remain consistent with the underlying causal relations.

# 4 EVALUATIONS

**Overview.** In this section, we conduct a systematic evaluation on our benchmark Causal3D, focusing on three major causal tasks as aforementioned. For each task, we select appropriate models to assess their performance across diverse scenes. For causal discovery from tabular data, we evaluate traditional causal discovery methods alongside an LLM-based causal agent. For causal representation learning, we benchmark state-of-the-art methods to examine their ability to extract meaningful causal factors. Lastly, for causal discovery and intervention from few images, we test various VLMs with different prompts. Our experiments span multiple settings with comprehensive insights.

## 4.1 EXPERIMENT SETTINGS

**Data Preprocessing.** Both image and tabular data were meticulously prepared to ensure seamless compatibility with the evaluated models. Image data were resized and normalized as required, and tabular data were formatted to meet the input specifications of traditional causal discovery methods.

**Evaluation Metrics.** We mainly use two metrics to quantify the experimental results: **1) F1 Score** is used to evaluate causal discovery results, which represents the harmonic mean of precision and recall of discovered causal relations. **2) Accuracy** is used in causal intervention, which measures the fraction of consistent causal relations in the intervened image. Each experiment is repeated 10 times per setting, and the final results are obtained by averaging these results for robust evaluation.

## 4.2 CAUSAL DISCOVERY FROM TABULAR DATA

**Evaluated Methods.** We evaluate the performance of various causal discovery methods on tabular data, covering both traditional algorithms and emerging LLM-powered approaches. We assess 7 traditional methods: CAM, NoTears, DAG GNN, DiffAN, PC, SCORE (Rolland et al., 2022), and GraN DAG; 1 LLM-based method: Causal Copilot (Wang et al., 2024b). All experiments were conducted on an NVIDIA RTX 4090 GPU.

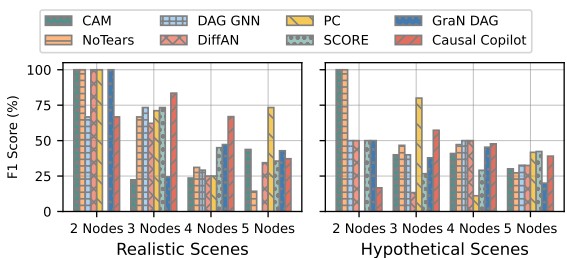

Figure 3: Results of different causal discovery methods from tabular data on realistic/hypothetical scenes.

**Results and Analysis.** The results are shown in Fig. 3. We test all methods on datasets representing both real and hypothetical scenes, categorized by causal graph complexity, ranging from 2- to 5-node structures. Each method takes the input of given tabular data and outputs a causal graph, and we compare it with the ground-truth causal graphs using the F1 score. The final evaluation metric is computed by averaging F1 scores within each node category. As shown in Fig. 3, although the performance of different methods varies within the same category, it can be observed in both realistic scenes and hypothetical scenes that there is a general downward trend in performance from 2-node to 5-node scenes. This aligns with our common sense and the laws of physics: the more variables there are in a scene, the more difficult it is to uncover the underlying rules.

## 4.3 CAUSAL REPRESENTATION LEARNING

**Evaluated Methods.** In this section, we evaluate causal representation learning models, including CausalVAE, DEAR, ICM-VAE (Komanduri et al., 2023), and CDG-VAE (An et al., 2023) on our dataset. The images and other information needed by models (e.g., CausalVAE requires annotations of causal variables) are taken as input. These methods are all based on the variational autoencoder (VAE) framework and aim to learn a low-dimensional representation composed of multiple disentangled yet causally related latent variables inside images.

**Evaluation Strategy.** Since many of these models require a (partial) causal graph as input or supervision, it would be unfair to directly evaluate them with regular causal discovery metrics we used in the previous subsection. Following the evaluation strategy of previous works (Yang et al., 2020; Shen et al., 2022; Komanduri et al., 2023; An et al., 2023), we apply interventions on each variable separately by modifying its corresponding causal representations, and then decode the modified latent representation back to generate an intervened image. By assessing whether the

causally related variables in the image change accordingly, we can evaluate whether the model has effectively learned the causal relations within our dataset.

Since these models only accept 2D images, we render the 3D scenes from a fixed viewpoint as inputs. For CausalVAE, DEAR, and ICM-VAE, we use *spring* and *seesaw* for evaluation. However, for CDG-VAE, which requires a structured mask to distinguish each object (i.e., variable), meaning that each object must move within a specific region, *spring* and *seesaw* do not satisfy this requirement. Therefore, we select *reflection* and *pendulum* for evaluation.

**Results and Analysis.** Intervention results are showcased in Fig. 4. For each given original image, we select a pair of "cause" and "result" variables to intervene with do-operation. For *spring*, the cause is the weight of the wooden block, and the result is the spring's deformation. We first apply an intervention on the weight of the wooden block (cause), with the

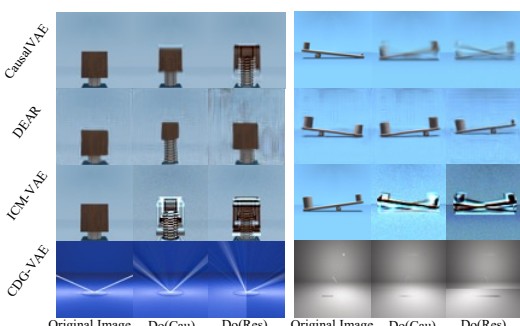

Figure 4: Examples of the 4 causal representation learning model results. In each scene, the 3 columns show: 1) original images, 2) Do(Cau): results after intervening on a "cause" variable, and 3) Do(Res): after intervening on a "result" variable.

intervened image shown in the left-middle column. We observe that decreasing the block's weight leads to a corresponding decrease in the spring's deformation. This aligns with the causal relation, where the block's weight directly influences the spring's deformation. However, when we directly modify the spring's deformation (result), as shown in the left-last column, the block's weight remains unchanged. This confirms the unidirectional nature of causal relations, where the cause affects the effect, but not vice versa. The results for the *seesaw* scene exhibit a similar pattern. When we apply an intervention on the torque on the left side (cause), as shown in the right-middle column, the seesaw's direction (result) reverses accordingly. However, when we directly intervene on the seesaw's direction, the torque on both the left and right sides remains unchanged. Similar observations hold for *reflection*, where the incident light serves as the cause and the reflected light as the result, and for *pendulum*, where the pendulum's angle acts as the cause, influencing the position and length of its shadow.

Although these models may not perform optimally in certain scenes, exhibiting distortions in reconstructed images and incomplete disentanglement of attributes, they still capture some underlying causal relations. This indicates the rationality of our proposed benchmark and highlights the need for further research in achieving more effective causal representation learning from images.

## 4.4 CAUSAL DISCOVERY FROM FEW IMAGES

**Evaluated Methods.** Unlike traditional causal discovery methods that rely on tabular data or large-scale training images, we leverage pretrained VLMs to perform causal discovery using a small number of images. In this section we use ChatGPT[2], Gemini[3] and Claude[4] to discover causal relations by few image examples and textual prompts in real and hypothetical scenes. The VLMs are tasked with uncovering causal relations by generating adjacency matrices representing causal graphs. Each prompt specifies key variables in the scenes, explicitly guiding the models to infer causal structures. We use 4 different prompting strategies, as detailed in Tab. 5 in the Appendix 7.

**1) Basic Prompts**: General instructions that broadly guide the models to identify causal relations.

**2) Explicit Function Prompts**: The model is designated as a causal discovery expert to identify causal relationships among image variables.

**3) Chain of Thought (CoT)**: The model is prompted to reason through the causal discovery process step by step without any prior examples. This approach encourages structured reasoning and provides insights into how the model interprets causal relations.

---

[2]https://platform.openai.com/docs/api-reference/introduction

[3]https://ai.google.dev/api?lang=python

[4]https://www.anthropic.com/api

**4) Few-Shot**: The model is given three exemplar cases of causal discovery before performing the task, enabling it to generalize better by leveraging prior examples to improve causal relation identification. Performance is evaluated by comparing the generated adjacency matrices with the ground truth, providing a quantitative measure of the causal discovery task.

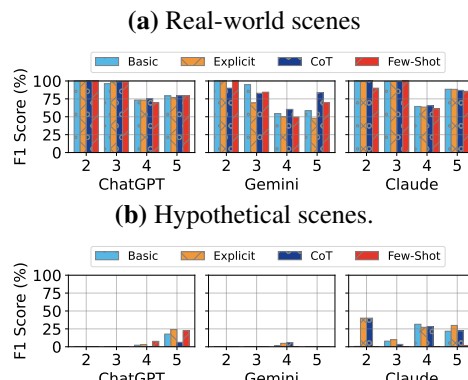

**(a)** Real-world scenes

**(b)** Hypothetical scenes.

**Results and Analysis**. Demonstrated in Fig. 5(a), models performed significantly better in real-world scenes (e.g., governed by Hooke's Law, magnetic fields and etc.) compared to hypothetical scenes, benefiting from prior physical knowledge embedded in their LLM components. However, as causal relations became more complex and involved more variables, performance declined, highlighting the challenge of uncovering causality in such scenes. In the hypothetical scenes, the results, as shown in Fig. 5(b), reveal poor performance. This suggests that when prior real-world knowledge is unavailable and only a limited number of images are provided, models consistently fail to uncover latent causal relations. Consequently, current closed-source VLMs have been shown to be unreliable for causal discovery in hypothetical settings.

Figure 5: Causal discovery results of VLMs in various scenes, averaged over datasets with 2–5 nodes in the causal graph.

**Views and Backgrounds.** CAUSAL3D provides multi-view images of the same scene, simulating real-world 3D environments with both virtual and realistic backgrounds. This enables in-depth analysis of how different viewpoints and background contexts affect causal discovery performance. Case studies from the Spring and Parabola scenes are shown in Fig. 6. Within the experiments, different 3D views affect the performance in causal discovery. Interestingly, our results reveal that intuitive views, such as a front view, are not always the most effective for uncovering latent causality. Different viewpoints can either increase or decrease task difficulty, with no single "golden standard" view for causal discovery. Additionally, when comparing multi-view and single-view inputs in the causal discovery task, multi-view performance varies depending on the scene (as shown in Fig. 9). In scenes with simple causal relations (e.g., a

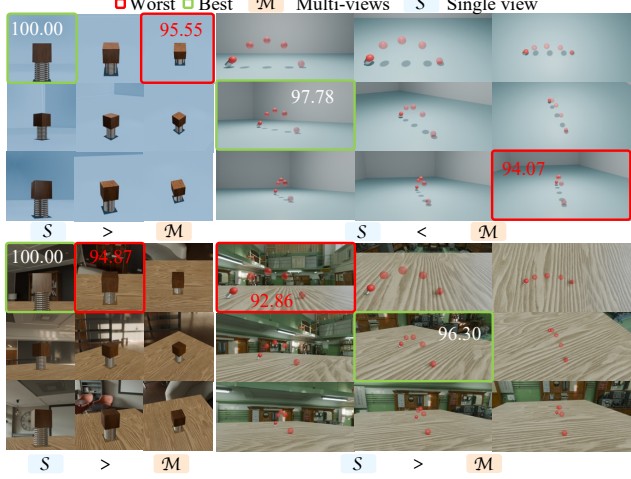

Figure 6: **Spring** and **Parabola** scene. Using the F1 score as metric, we assess VLM performance in causal discovery. The best and worst views are highlighted to demonstrate the impact of different views. To analyze the effect of multi-view vs. single-view images, we average the performance across 9 individual views in each scene and compare it with the overall multi-view performance. Details and concrete numerical results are provided in Appendix 7.2.

spring system with a linear relation among three variables), multi-view inputs tend to degrade model performance, possibly introducing unnecessary noise. Conversely, in more complex 3D scenes with intricate causal dependencies, like a parabola scene involving nonlinear relations among four variables in the virtual background, multi-view perspectives enhance inference accuracy, suggesting that additional viewpoints help capture richer causal structures in such settings.

Comparing virtual and realistic backgrounds, we find that realistic backgrounds introduce additional noise, making causal discovery more challenging. Our experiments indicate that even when models leverage prior real-world knowledge encoded in LLMs, the presence of realistic background information increases task complexity and negatively impacts inference performance (quantitative results shown in Fig. 37 in Appendix 7).

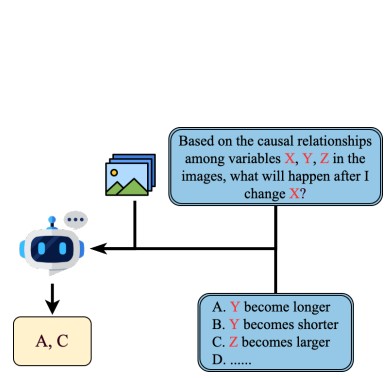

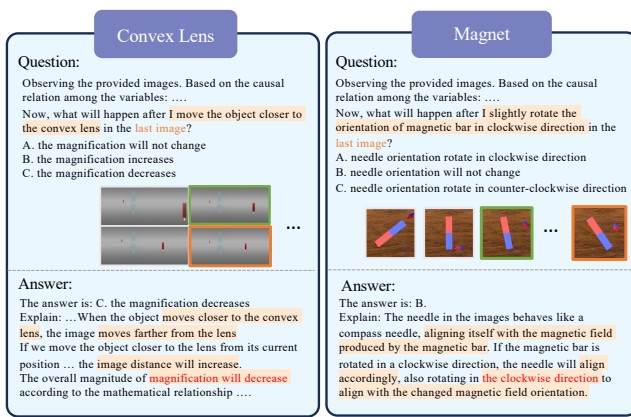

Figure 7: Example of intervention prompts for trained VLMs.

Figure 8: Case studies of failures. VLMs fail to grasp causality within physically consistent scenes for intervention tasks.

### 4.5 CAUSAL INTERVENTION IN VLMS

**Evaluated Methods.** To further assess the ability to learn causality from images, causal intervention serves as a crucial step beyond causal discovery. In this experiment, we leverage VLMs in interventions. This task is formulated as a multiple-choice problem, where the model must identify the correct outcomes based on induced causal changes. The detailed inference pipeline is illustrated in Fig. 7. In this setup, VLMs are provided with a few images along with questions about the variables present in the scene. By posing queries such as "*What will happen after changing variable X?*", we expect the model to return the correct answer based on the causal relations depicted in the given images. This evaluation determines whether VLMs genuinely comprehend and reason causal relations rather than relying solely on statistical patterns. In this experiment, intervention performance is measured by model selection accuracy. We evaluate 3 popular commercial VLMs on this task: ChatGPT-4o, Gemini-1.5-Pro, and Claude-3.5-Haiku[5].

**Results and Analysis.** As shown in Tab. 2, our experiments indicate that current VLMs struggle to handle complex rules, e.g. magnetic fields, in causal intervention tasks. We found that the models' inferences are dominated by the prior knowledge embedded in the LLMs, while visual cues—despite containing the ground truth—are largely ignored (see Fig. 8).

Table 2: Causal intervention in VLMs: Evaluation using three real-world rules, with accuracy (%) reported.

| Models | Reflection | Lens | Mag. Field |
|--------|-----------|--------|------------|
| ChatGPT | 96.67 | 100.00 | 50.00 |
| Gemini | 100.00 | 100.00 | 0.00 |
| Claude | 100.00 | 96.67 | 13.33 |

## 5 CONCLUSION

CAUSAL3D is a comprehensive benchmark designed to evaluate causal reasoning in visual AI, including causal tasks of discovery, disentanglement, and intervention. Our dataset integrates structured causal graphs with corresponding 3D visual representations, providing a rigorous assessment framework across diverse physical and hypothetical scenes. Experimental results demonstrate that: 1) the performance of current causal discovery algorithms decreases as the increasing of the complexity of causal structures; 2) for causal representation learning, the current state-of-arts can not well handle our realistic and diverse 3D scenes; 3) commercial VLMs struggle with causal inference based on visual cues and complex scenarios, particularly in hypothetical settings where prior knowledge is absent. CAUSAL3D serves as a critical step toward bridging the gap between traditional causal research and computer vision, enabling a more comprehensive and fine-grained evaluation of causal inference. We envision CAUSAL3D as a foundation for future research, fostering advancements in causal-aware AI models and driving progress toward more reliable and interpretable machine intelligence.

---

[5]Model versions are those publicly available as of January 2025.

## ETHICS STATEMENT

This research adheres to the ICLR Code of Ethics. The study does not involve human subjects, personally identifiable information, or sensitive data. All datasets used are publicly available, and appropriate steps were taken to ensure compliance with privacy, fairness, and research integrity standards. No conflicts of interest or ethical concerns are anticipated.

## REPRODUCIBILITY STATEMENT

We conducted extensive experiments with multiple random seeds to ensure the robustness and reproducibility of our results. Detailed descriptions of model architectures, training procedures, and hyperparameters closely follow those reported in the original baseline papers. The data generation code is submitted in a zip file as supplementary materials.

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

# APPENDIX

In the appendix, we will provide the statistics of Causal3D and supplement the scenes and their data details that were not presented in the main text. Additionally, we will also display the remaining experiment setups and results.

## 6 DATASET DETAILS AND ADDITIONAL SCENES

Causal3D is annotated with causal graphs that vary in node count and graph structure. For a global overview of Causal3D, we summarize the **number of settings** and the **number of causal structures** under different numbers of variables in each scene in Tab. 3a and Tab. 3b.

Fig. 9, Fig. 10, and Fig. 11 show the dataset details of different scenes, including the following information: name of scenes, the number of causal variables (i.e., nodes in causal graph), the relation types (linear or non-linear), causal graphs, and structural equations. For realistic scenes, we also add a brief description. Furthermore, we showcase all the scenes by randomly sampling 2D images from different viewpoints and surroundings for illustration (shown from Fig. 12 to Fig. 36). For notational simplicity, we omit the exogenous noise variables in the structural equations, all of which follow uniform distributions. For example, in the Reflection scene, the expression $A = B$ is a shorthand for the structural equation $B = A + \epsilon$, where $\epsilon$ is an independent uniform noise term. This non-Gaussianity ensures the identifiability of the causal direction $A \to B$.

| Scene | Description | Causal Graph | Structure Equation |
|---|---|---|---|
| Reflection, Linear | *Light reflects off a mirror.* | A $\to$ B | A: Incident light. B: Outgoing light. | $A = B$ |
| Seesaw, Non-Linear | *A seesaw with a cylinder on each end.* | A, B, C | A: Left torque. B: Right torque. C: Seesaw tilt direction. | $C = sign(A - B)$ |
| Convex Lens, Non-Linear | *A candle and its image formed by a convex lens.* | A, B, C | A: Distance from lens to the object. B: Distance from the lens to the image. C: Magnification factor. | $\frac{1}{f} = \frac{1}{A} + \frac{1}{B}$   $f$ is the focal length, which is a constant.  $C = -\frac{A}{B}$ |
| Magnet, Non-Linear | *A magnet and a needle displaying its magnetic field.* | A, B, C, D | A: Rotation angle of the bar magnet. B: The x-coordinate of the magnetic needle. C: The y-coordinate of the magnetic needle. D: Orientation of the magnetic field at the needle. | $D = \frac{\mu_0}{4\pi}\left(\frac{3\overrightarrow{(B,C)}\vec{A}(B,C)}{\overrightarrow{(B,C)}^5} - \frac{\vec{A}}{\overrightarrow{(B,C)}^3}\right)$ |
| Pendulum, Non-Linear | *A light source, a pendulum, and its shadow.* | A, B, C, D, E | A: Light position. B: Pendulum angle. C: Pendulum length. D: Shadow middle point position. E: Shadow length. | $D = \frac{1}{2}\left(-\frac{y_l x_p + y_l C sinB + ACcosB - y_p A}{y_p - CcosB - y_l} - \frac{y_l x_p - y_p A}{y_p - y_l}\right)$  $E = -\frac{y_l x_p + y_l C sinB + ACcosB - y_p A}{y_p - CcosB - y_l} + -\frac{y_l x_p + y_l C sinB + ACcosB - y_p A}{y_p - CcosB - y_l}$  $(x_p, y_p)$ is the coordinate of the pendulum end point.  $y_l$ is the y-coordinate of the light. |

Figure 9: Data details of realistic scenes (as a supplement to Fig. 1).

| Scene | Causal Graph | Structure Equation |
|---|---|---|
| 2 Variables, Linear | A → B | A: Volume of the ball. B: Volume of the cube. | $B = 1.5A$ |
| 2 Variables, Non-Linear | A → B | A: Volume of the ball. B: Volume of the cube. | $B = \cos(A)$ |
| 3 Variables, Fully Connected, Linear | A, B, C (triangle) | A: Volume of the ball. B: Height of the cuboid. C: Base area of the cone. | $B = 4A$
$C = -10A + 10B$ |
| 3 Variables, V-Structure, Linear | A, B, C | A: Volume of the ball. B: Height of the cuboid. C: Base area of the cone. | $C = 0.4A + 0.7B$ |
| 3 Variables, V-Structure, Non-Linear | A, B, C | A: Volume of the ball. B: Height of the cuboid. C: Base area of the cone. | $C = \tan(A) + 0.7B$ |

Figure 10: Data details of hypothetical scenes (2 variables and 3 variables).

| # Nodes | 2 | 3 | 4 | 5 |
|---|---|---|---|---|
| **Realistic** | 2 | 2 | 2 | 2 |
| **Hypothetical** | 2 | 3 | 3 | 3 |

(a) Number of settings per node count.

| # Nodes | 2 | 3 | 4 | 5 |
|---|---|---|---|---|
| **Realistic** | 1 | 2 | 2 | 2 |
| **Hypothetical** | 1 | 2 | 2 | 2 |

(b) Number of causal structures per node count.

Table 3: The basic statistic of Causal3D

| Scene | Causal Graph | | Structure Equation |
|---|---|---|---|
| 4 Variables, No V-structure Linear |  | A: Volume of the ball.
B: Volume of the cube.
C: Base area of the cuboid.
D: Base area of the cone. | $A = 0.5D$
$B = 0.3A$
$C = 0.4A + 0.6B + 0.9B$ |
| 4 Variables, V-Structure, Linear |  | A: Volume of the ball.
B: Volume of the cube.
C: Base area of the cuboid.
D: Base area of the cone. | $C = 0.3A + 0.7B$
$D = 0.4C$ |
| 4 Variables, V-Structure, Non-Linear |  | A: Volume of the ball.
B: Volume of the cube.
C: Base area of the cuboid.
D: Base area of the cone. | $C = 50\sin(A) + 20B$
$D = 1100\cos(C)$ |
| 5 Variables, No V-structure Linear |  | A: Volume of the ball.
B: Volume of the cube.
C: Base area of the cuboid.
D: Base area of the cone.
E: Height of the cone. | $B = 0.01A$
$C = -0.01A + 16B$
$D = 1.2C$
$E = 5A - 0.5C + 2D$ |
| 5 Variables, V-Structure, Linear |  | A: Volume of the ball.
B: Volume of the cube.
C: Base area of the cuboid.
D: Base area of the cone.
E: Height of the cone. | $B = 0.045D$
$C = 0.03A + 10B$
$E = 0.01C + 0.02D$ |
| 5 Variables, V-Structure, Non-Linear |  | A: Volume of the ball.
B: Volume of the cube.
C: Base area of the cuboid.
D: Base area of the cone.
E: Height of the cone. | $B = 60\sin(D)$
$C = 400\cos(A) + 20B$
$E = 35\tan(C) + 0.1D$ |

Figure 11: Data details of hypothetical scenes (4 variables and 5 variables).

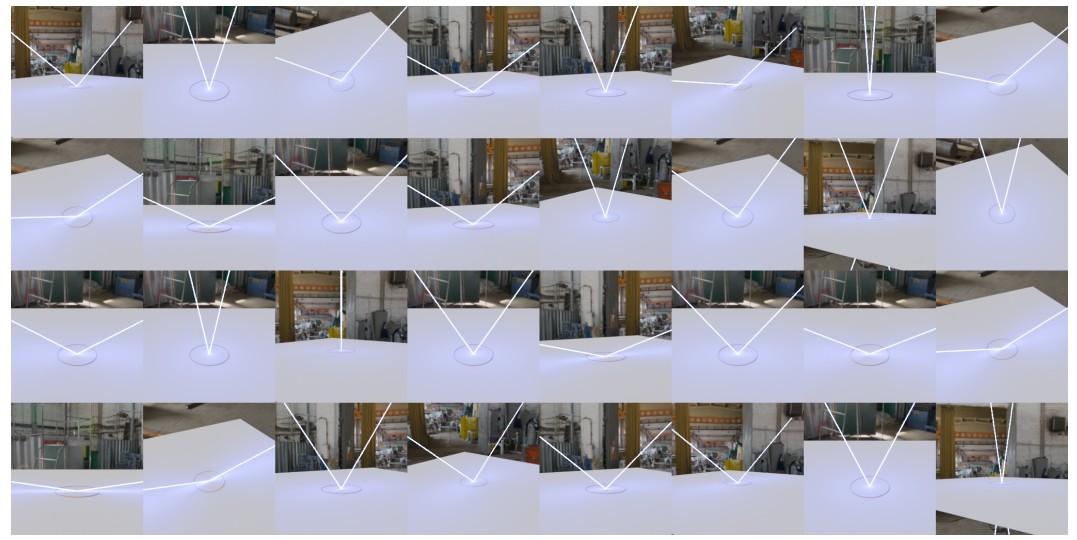

Figure 12: Reflection (Real Background).

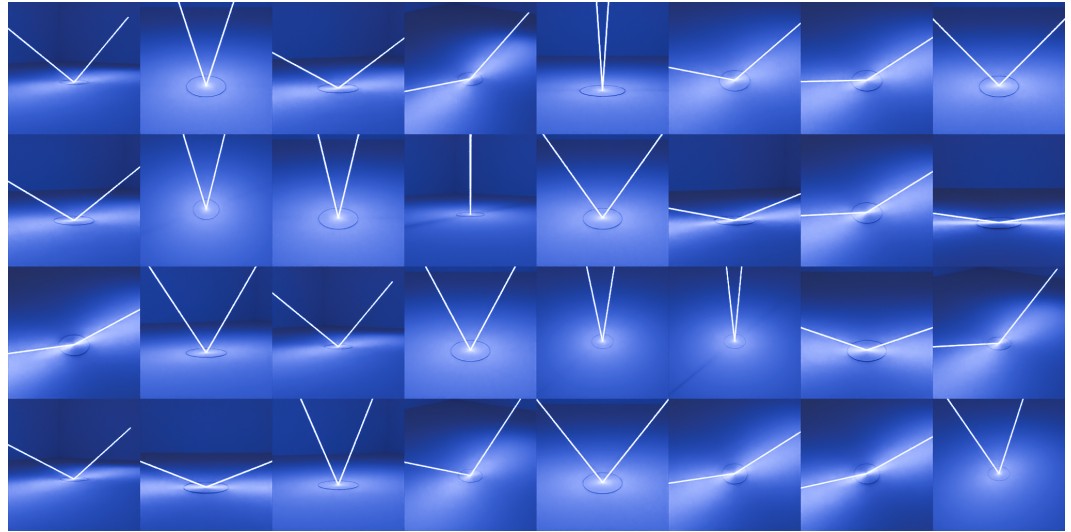

Figure 13: Reflection (Virtual Background).

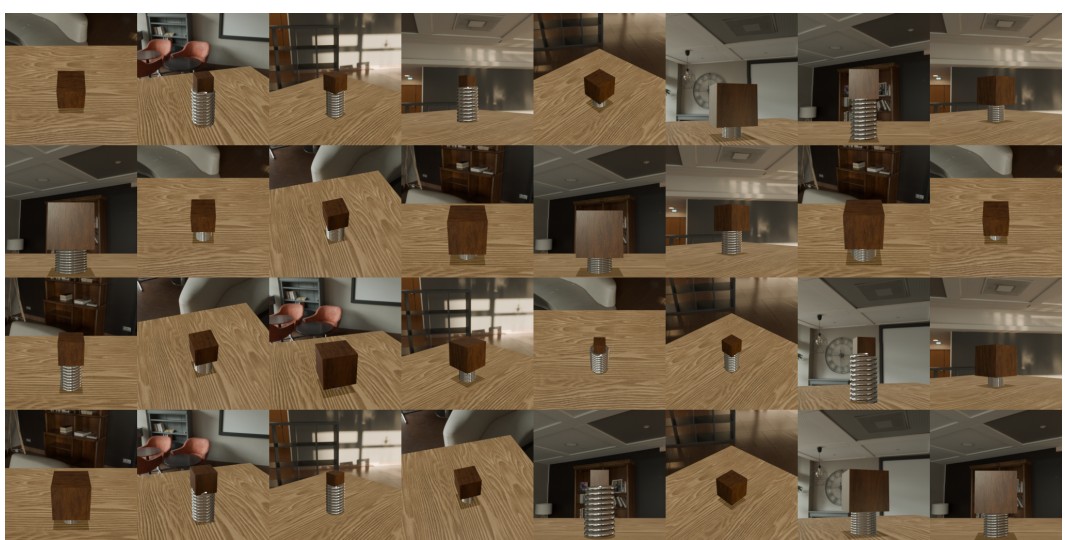

Figure 14: Spring (Real Background).

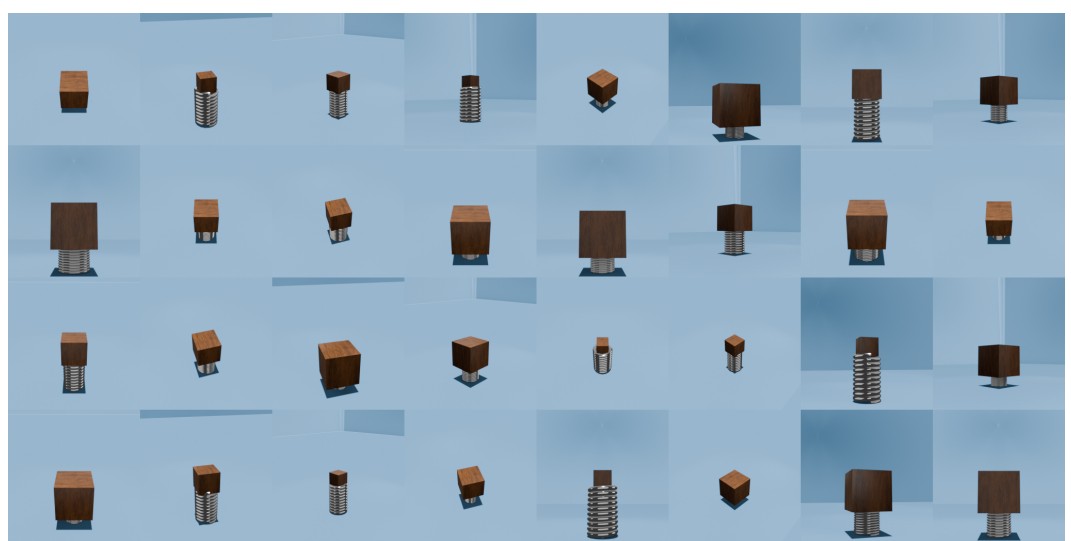

Figure 15: Spring (Virtual Background).

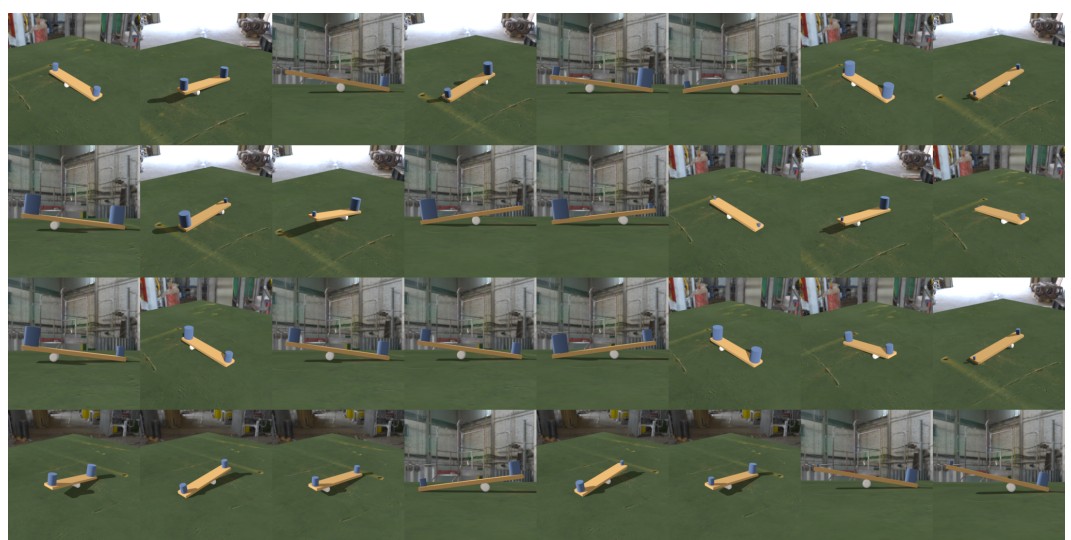

Figure 16: Seesaw (Real Background).

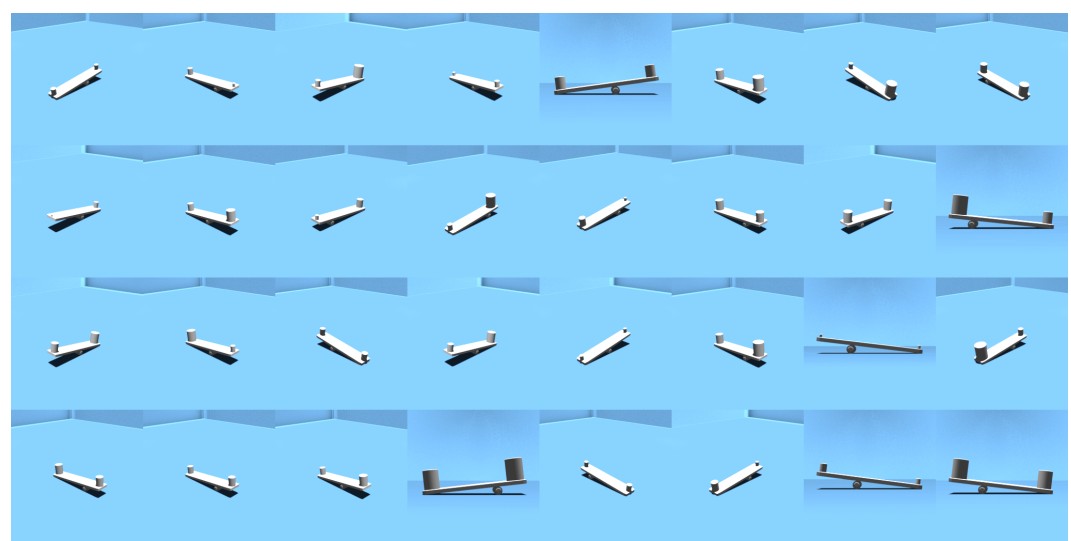

Figure 17: Seesaw (Virtual Background).

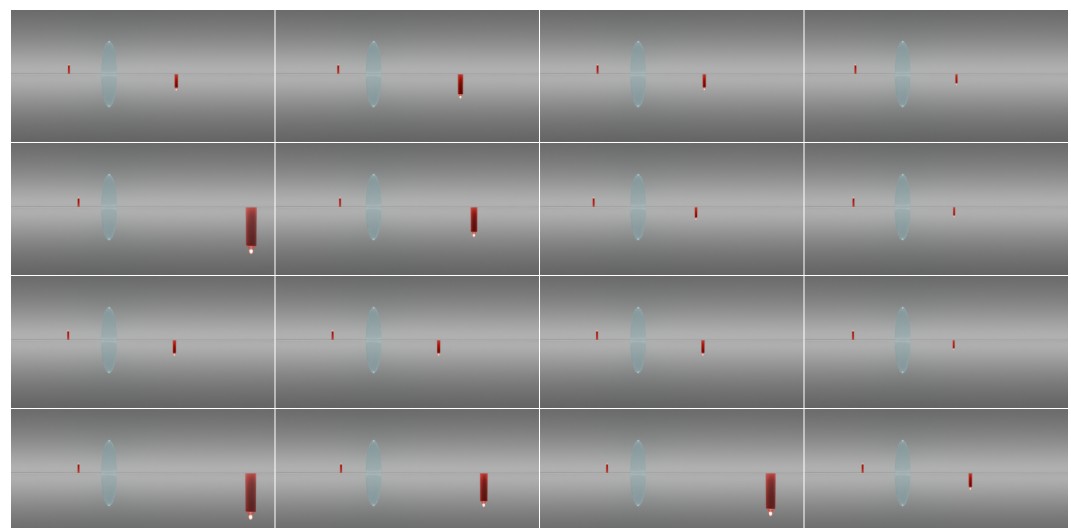

Figure 18: Convex Lens.

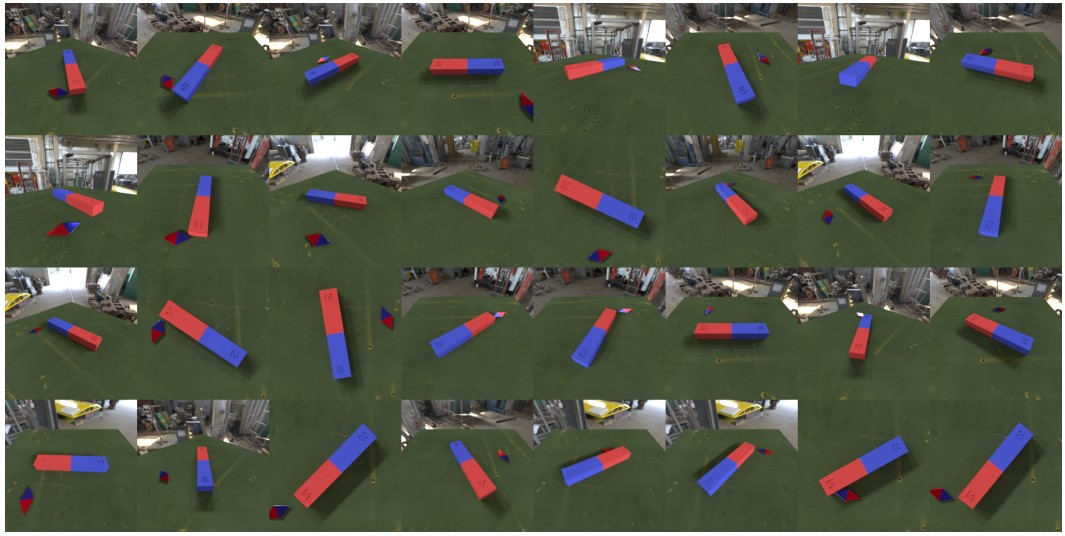

Figure 19: Magnet (Real Background).

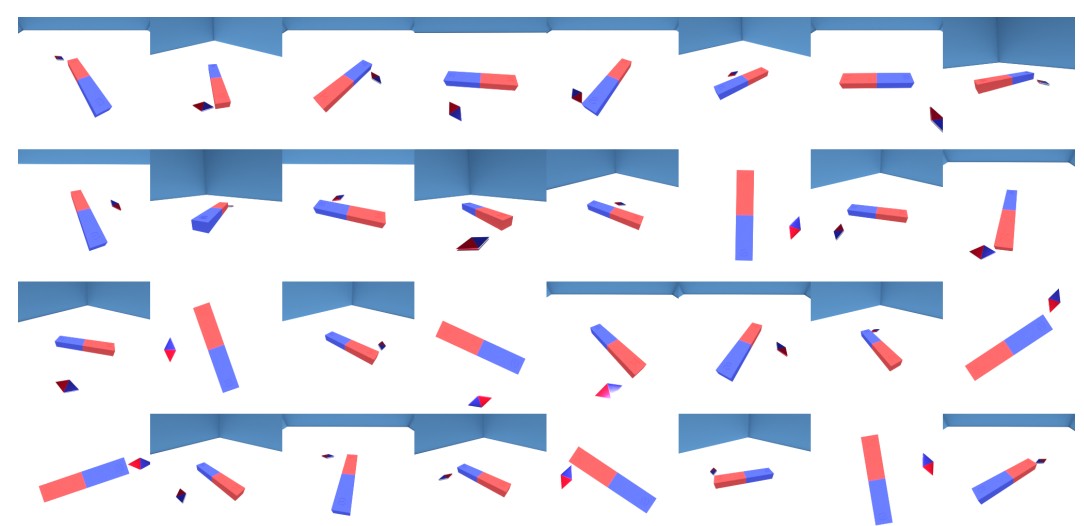

Figure 20: Magnet (Virtual Background).

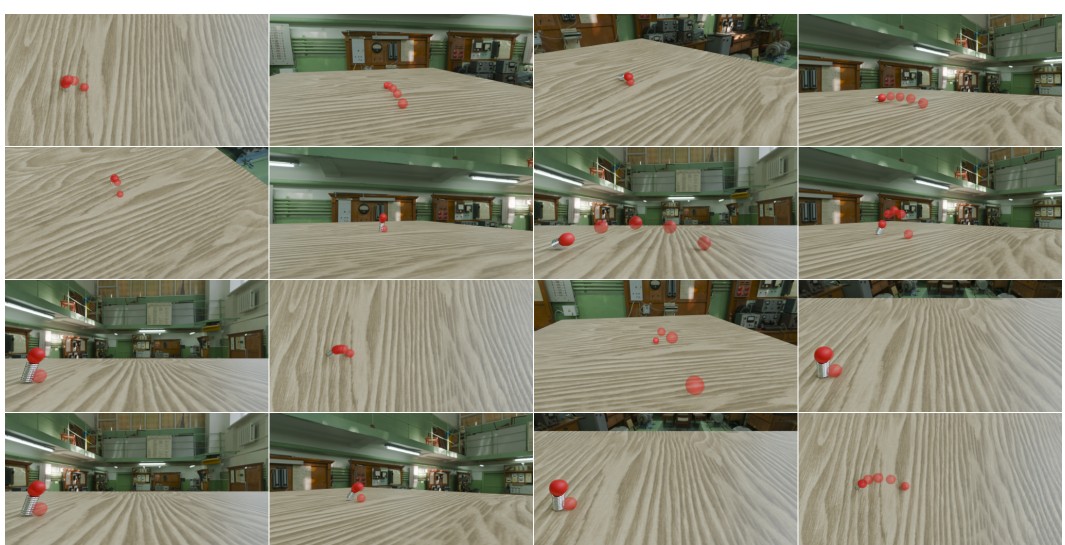

Figure 21: Parabola (Real Background).

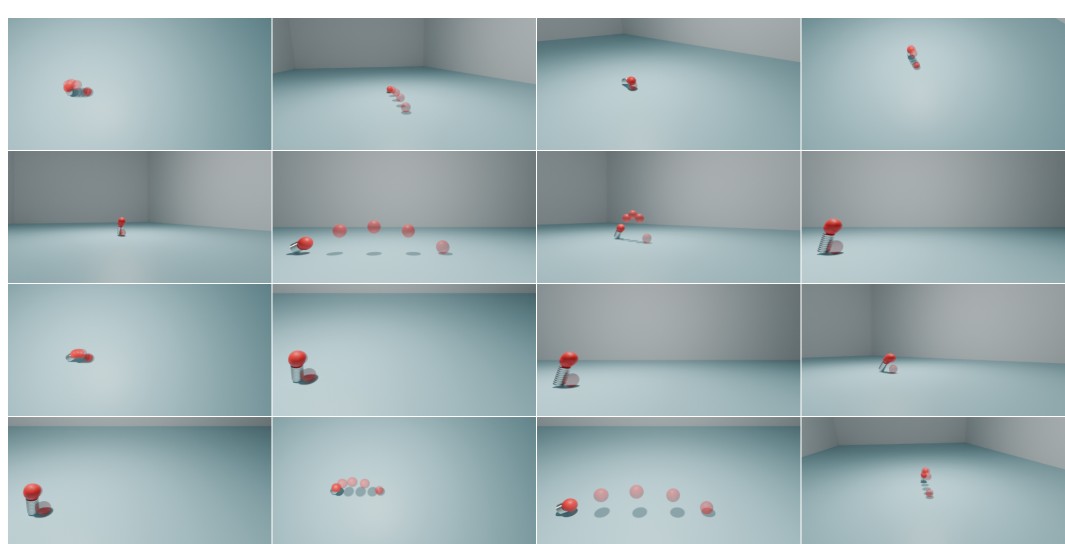

Figure 22: Parabola (Virtual Background).

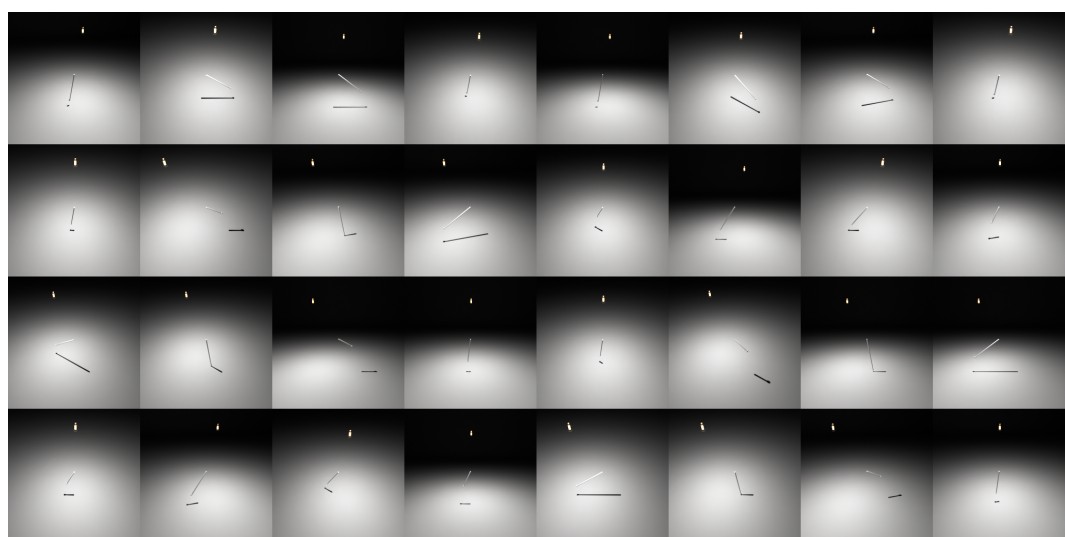

Figure 23: Pendulum.

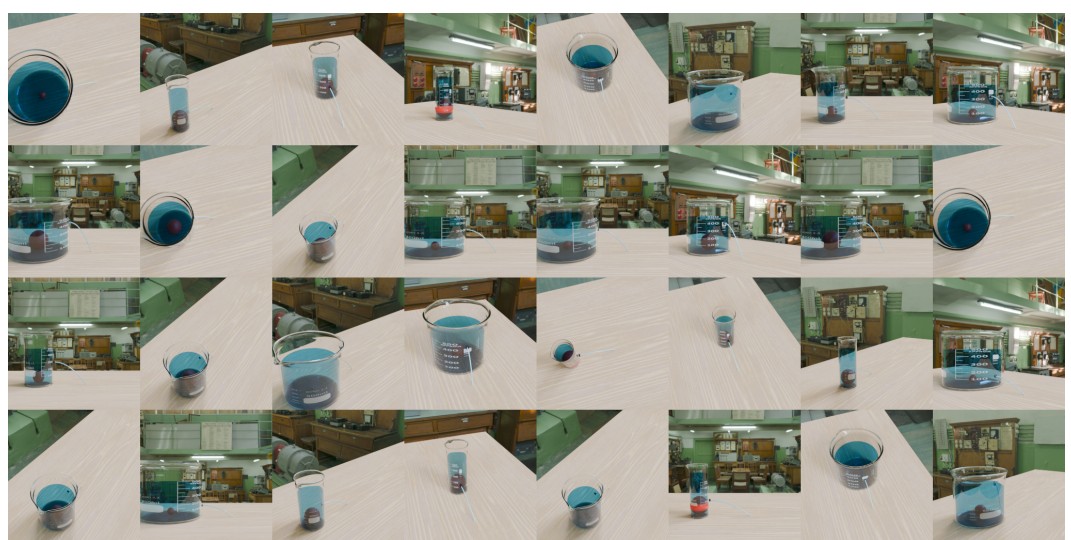

Figure 24: Water Flow (Real Background).

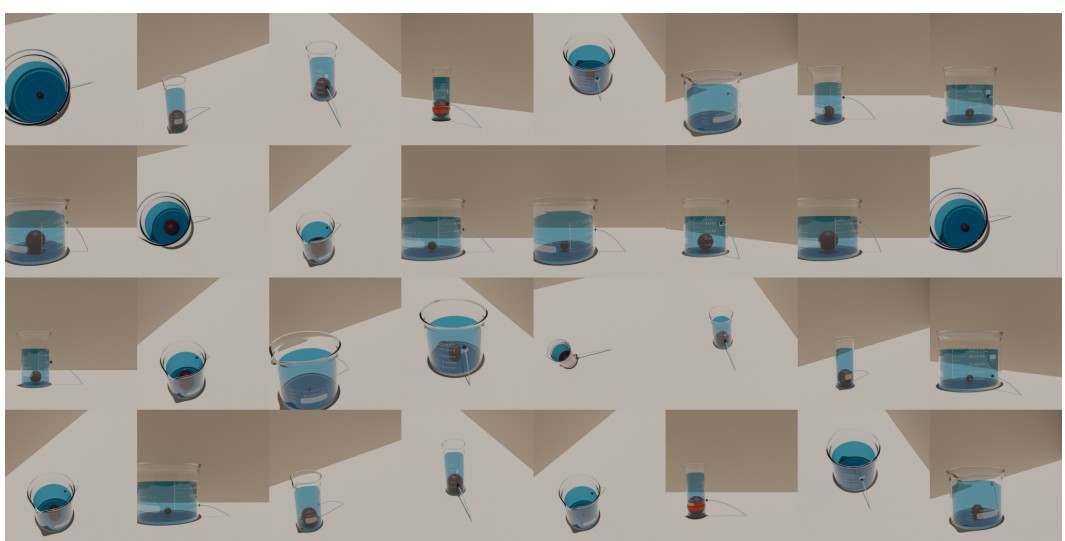

Figure 25: Water Flow (Virtual Background).

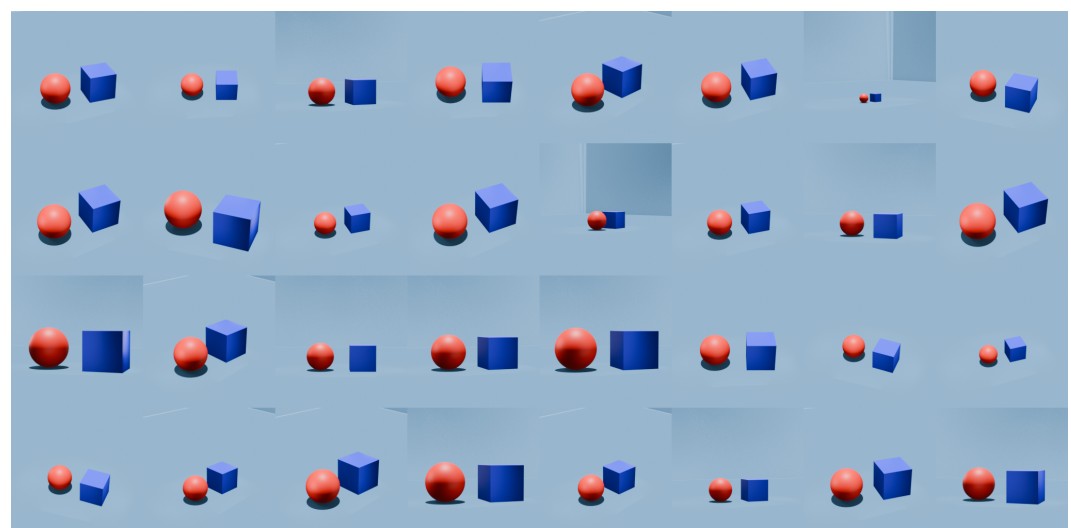

Figure 26: Hypothetical Scene (2 Variables, Linear).

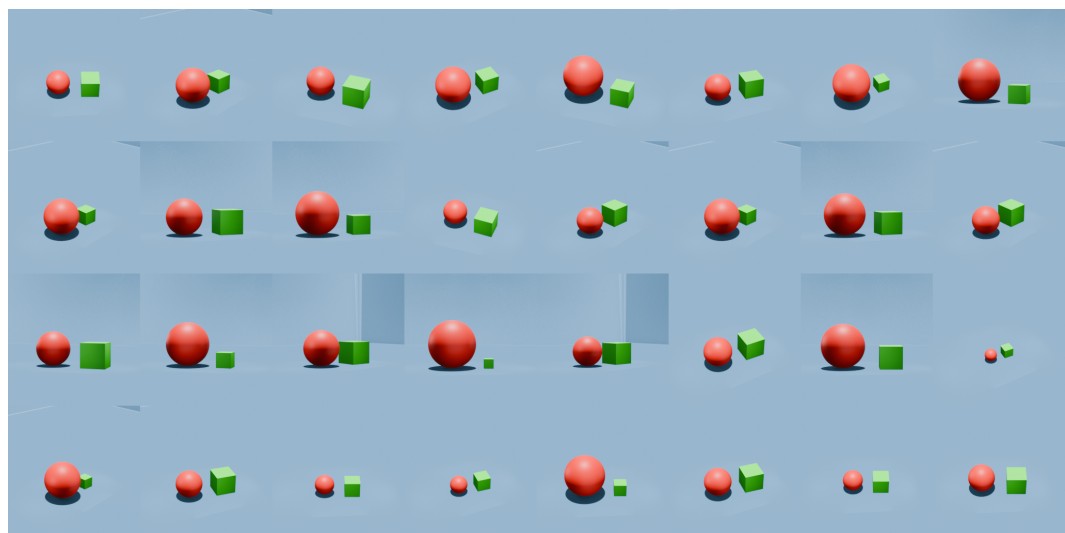

Figure 27: Hypothetical Scene (2 Variables, Non-Linear).

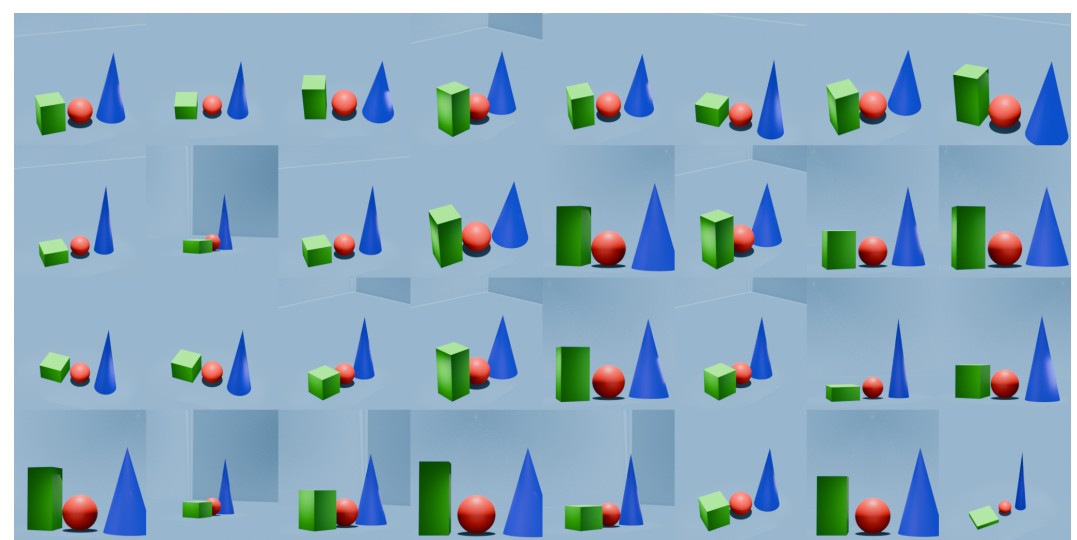

Figure 28: Hypothetical Scene (3 Variables, Linear, Fully-Connected).

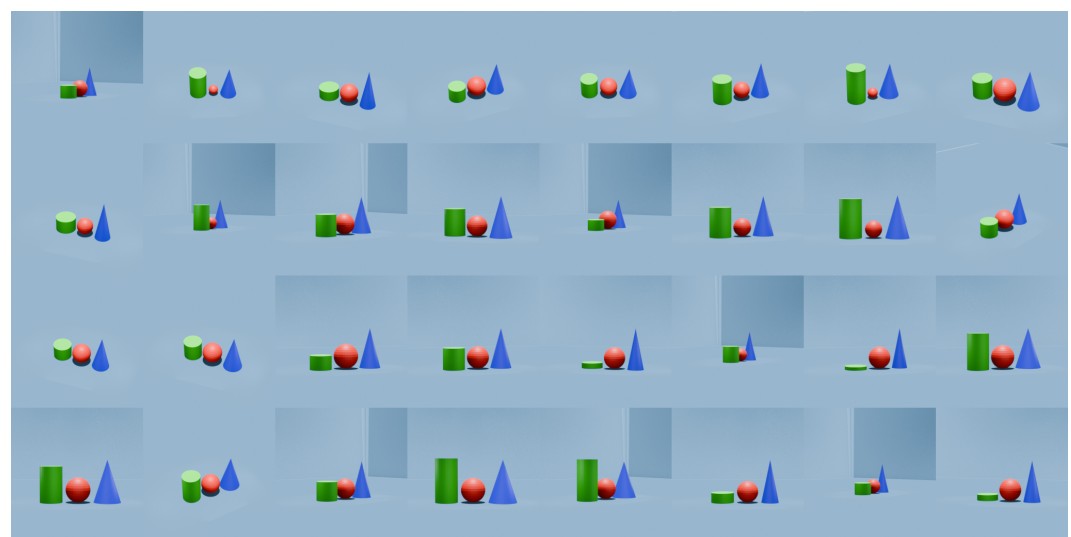

Figure 29: Hypothetical Scene (3 Variables, Linear, V-Structure).

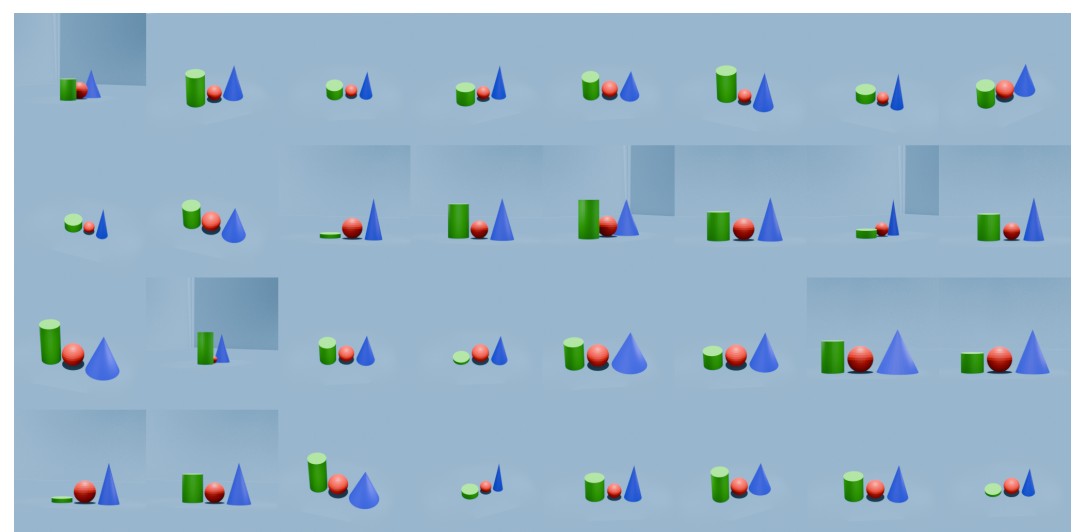

Figure 30: Hypothetical Scene (3 Variables, Non-Linear, V-Structure).

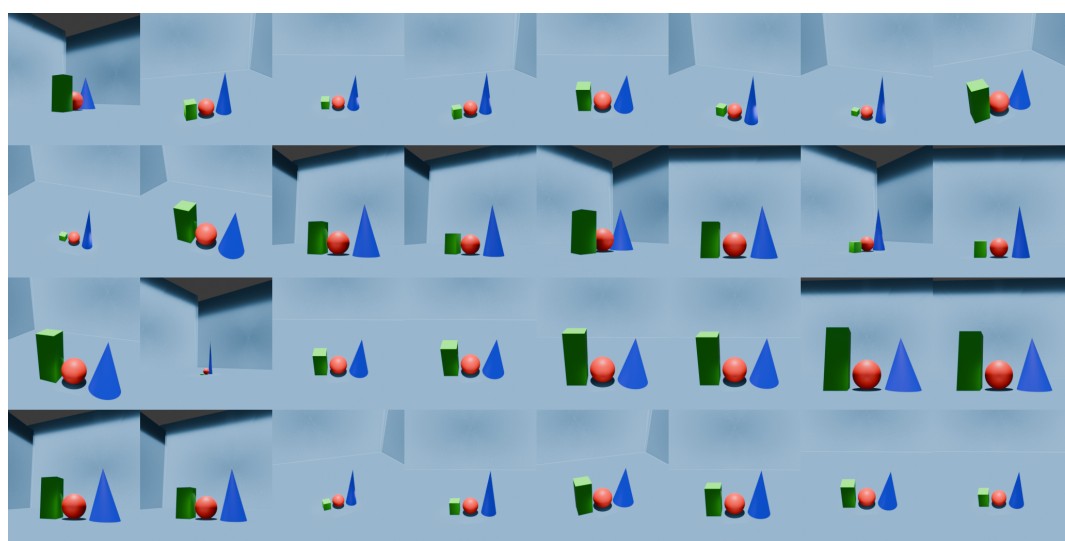

Figure 31: Hypothetical Scene (4 Variables, Linear, Fully-Connected).

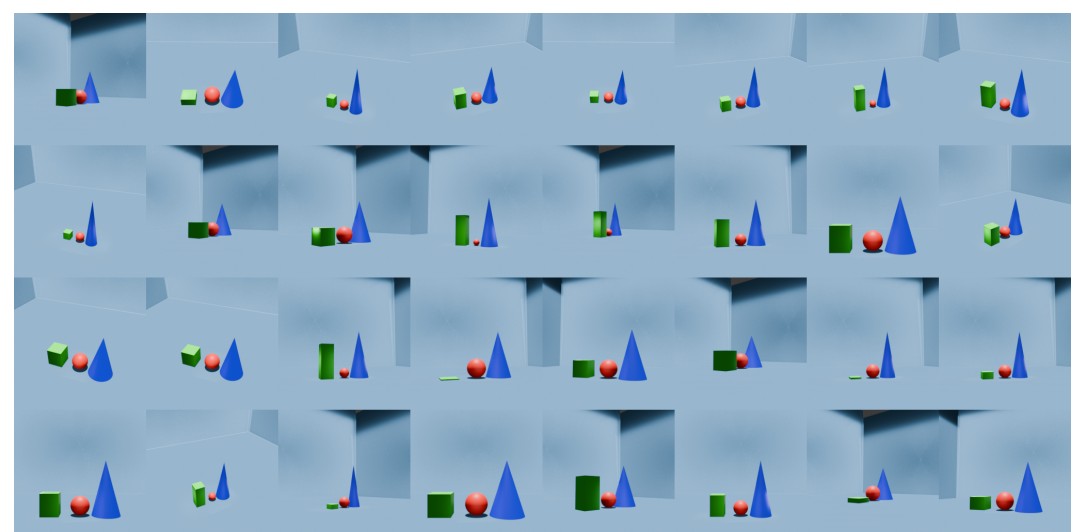

Figure 32: Hypothetical Scene (4 Variables, Linear, V-Structure).

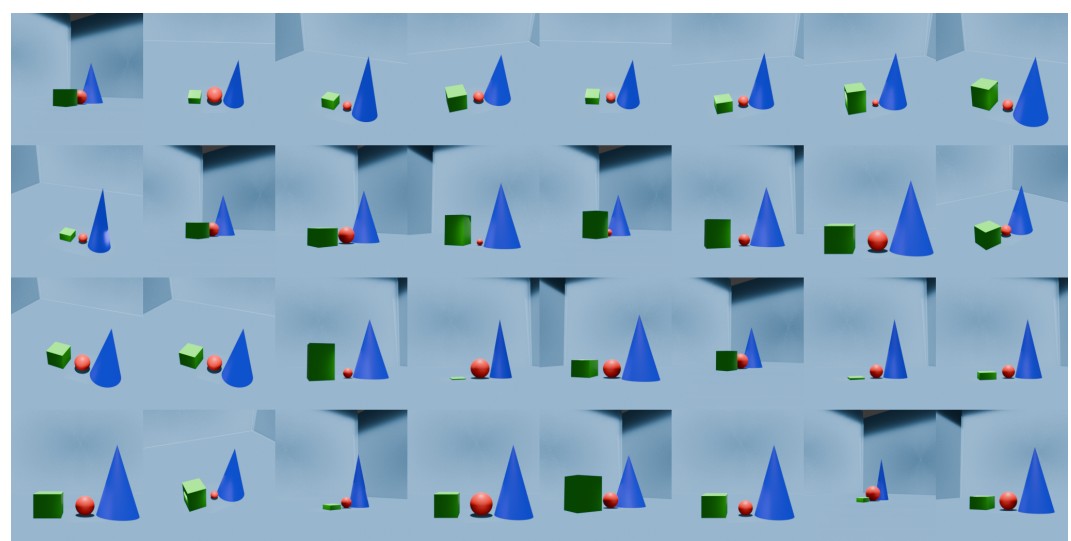

Figure 33: Hypothetical Scene (4 Variables, Non-Linear, V-Structure).

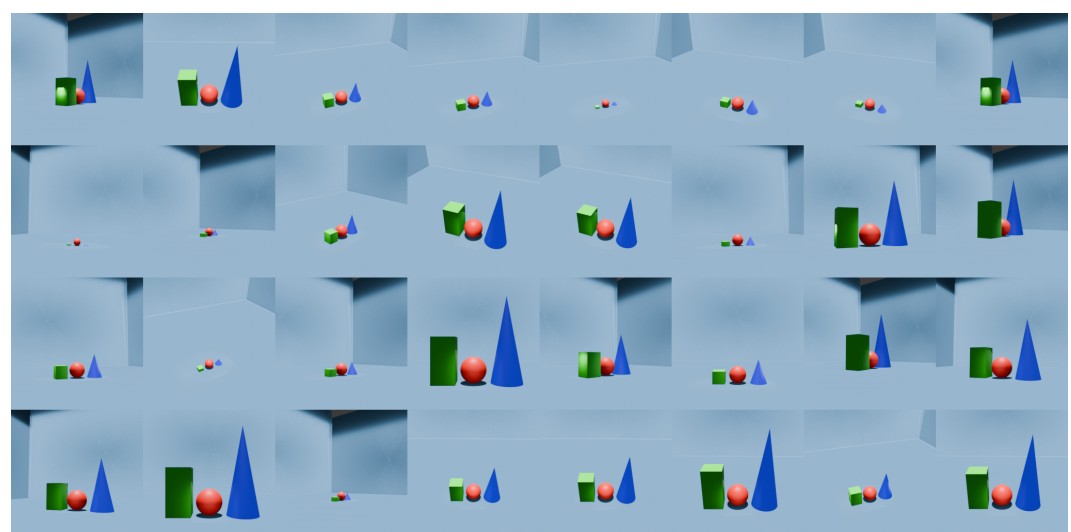

Figure 34: Hypothetical Scene (5 Variables, Linear, Fully-Connected).

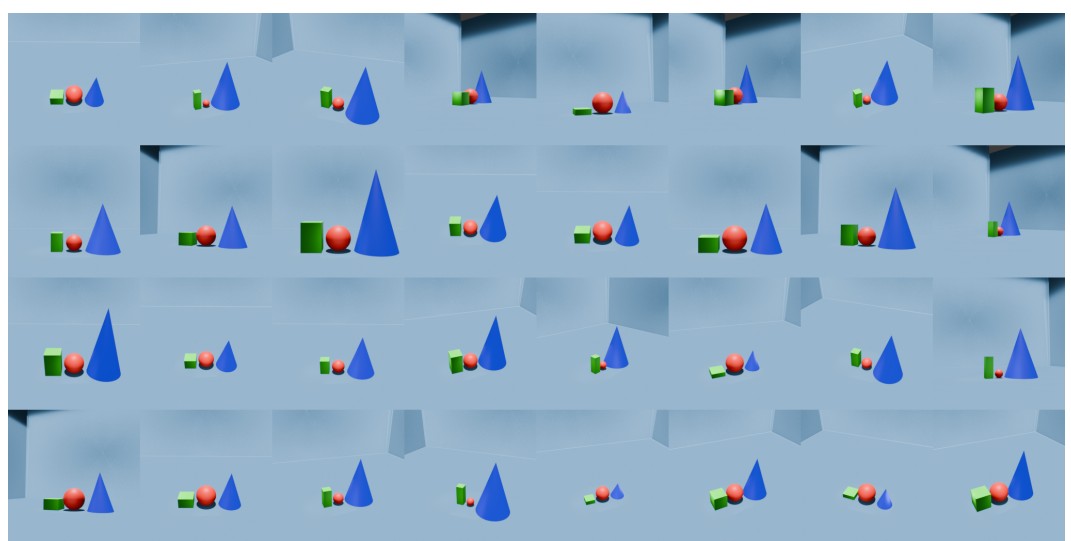

Figure 35: Hypothetical Scene (5 Variables, Linear, V-Structure).

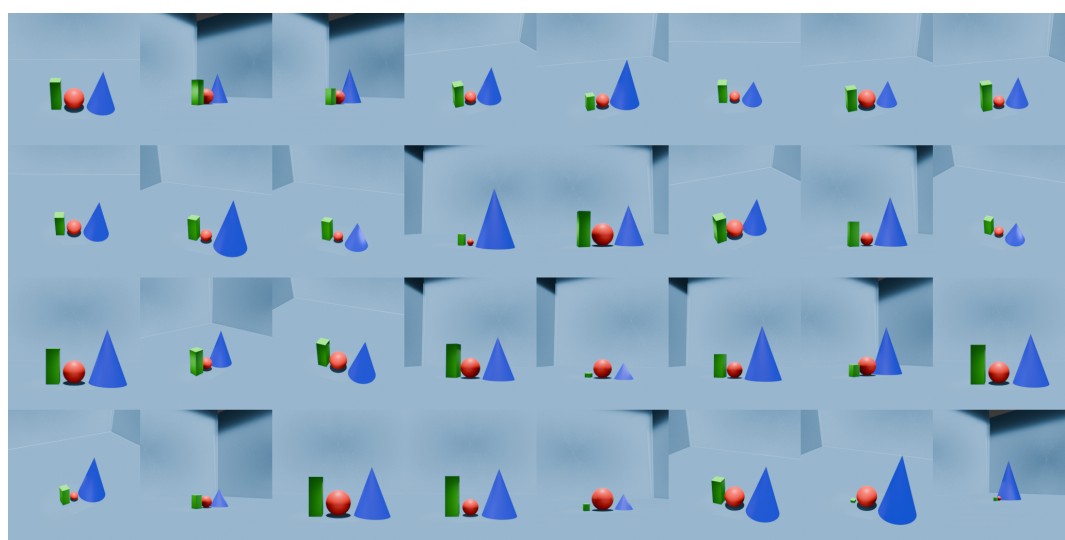

Figure 36: Hypothetical Scene (5 Variables, Non-Linear, V-Structure).

# 7 EXPERIMENT DETAILS

In this section, we record the details in the experiments in the main paper.

## 7.1 DIFFERENT PROMPTING STRATEGIES

In the experiment of VLMs, we present four detailed prompt strategies for performing causal discovery tasks (see Tab. 5).

## 7.2 EXPERIMENTAL RESULT OF THE IMPACT OF VIEWS AND BACKGOUNDS

In the main paper, we evaluate the impact of different views and backgrounds on causal discovery with VLMs. As shown in Fig.37, we use multiple images with both realistic and virtual backgrounds in the experiments. Tab. 4 reports the concrete numerical results for each setting, which support our claims in Sec. 4.4.

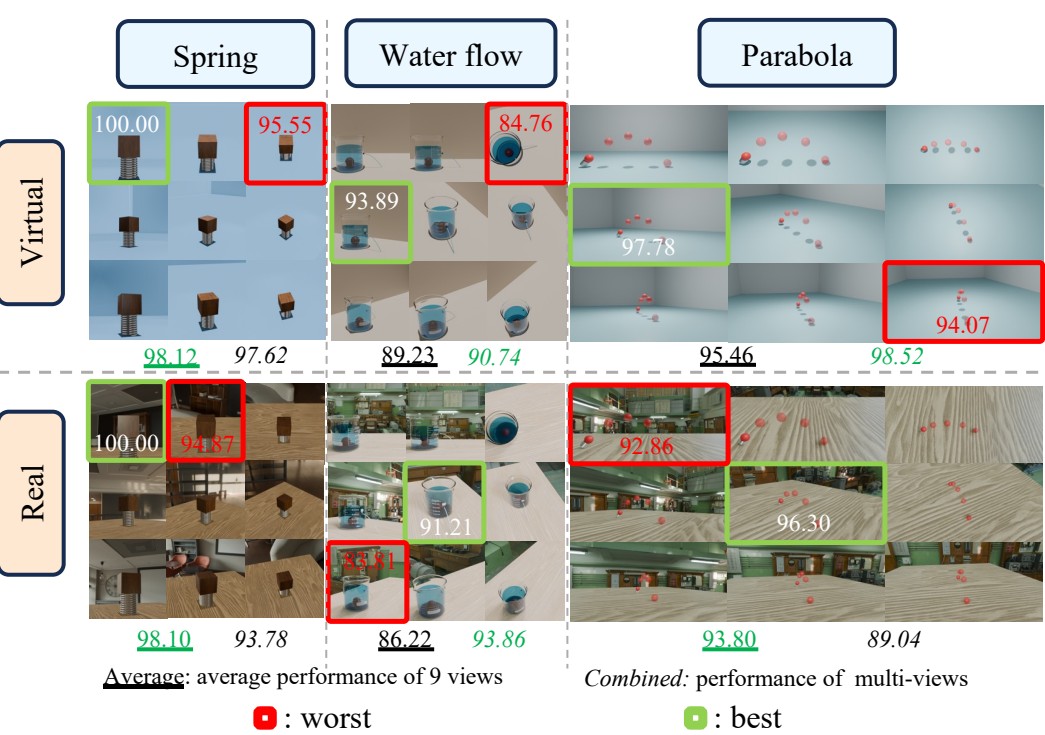

Figure 37: Performance Comparison in 3D scenes: selecting 3 scenes for case studies: Spring, Water flow, and Parabola. Using F1 score as the evaluation metric, we assess inference performance in the causal discovery task. The best and worst views are highlighted to demonstrate the impact of different perspectives. To analyze the effect of multi-view vs. single-view inputs, we average the performance across 9 individual views and compare it with the overall multi-view performance, highlighting the better results in green.

# 8 BROADER IMPACT

CAUSAL3D advances the integration of causal reasoning in computer vision, contributing to more robust, interpretable, and generalizable AI systems. By introducing a structured benchmark and systematically evaluating state-of-the-art methods, our research provides valuable insights into the challenges and opportunities of causal learning in visual data. The proposed benchmark fosters interdisciplinary collaboration, bridging causal inference, computer vision, and machine learning communities. It serves as a foundation for future research, enabling the development of models that can better generalize across domains, adapt to distribution shifts, and provide meaningful

| 100 | 100 | 95.56 | 86.87 | 90.07 | 84.76 | 94.09 | 94.42 | 95.56 |
|---|---|---|---|---|---|---|---|---|
| 94.87 | 100 | 100 | 93.89 | 91.39 | 87.95 | 97.78 | 95.56 | 96.83 |
| 97.62 | 97.78 | 97.22 | 88.79 | 91.10 | 88.22 | 95.11 | 95.70 | 94.07 |
| 100 | 94.87 | 97.78 | 87.53 | 83.97 | 85.93 | 92.86 | 93.00 | 94.56 |
| 100 | 97.44 | 97.62 | 81.71 | 91.21 | 83.82 | 93.11 | 96.30 | 94.04 |
| 95.24 | 100 | 100 | 86.99 | 88.60 | 86.22 | 93.07 | 93.45 | 93.80 |

Table 4: Numerical results corresponding to Fig. 37. The table layout matches Fig. 37, and each value indicates performance under a specific view.

explanations. Furthermore, by improving causal understanding in vision tasks, this work has potential applications in fields such as healthcare, autonomous systems, and scientific discovery, where reliability and transparency are essential. While our evaluation framework is based on the authors' consensus, we encourage community discussions to refine causal reasoning criteria and enhance benchmarking standards. We will release evaluation scripts to support innovation and aid the development of new methodologies. Additionally, we emphasize the responsible use of CAUSAL3D and strictly prohibit any form of data leakage or test set optimization to maintain fairness and integrity in evaluation. Our work does not raise any ethical concerns that require disclosure.

## 9 LIMITATIONS

Although CAUSAL3D is comprehensive, there remains room for improvement. First, the current benchmark is constructed solely from observational data; incorporating interventional data in the future would enable user interaction and support more interactive causal evaluation. Second, the dataset complexity could be further enriched by introducing more nodes in the causal graphs and incorporating finer-grained visual details, such as textures of objects.

## 10 THE USE OF LARGE LANGUAGE MODELS (LLMS)

We used ChatGPT, Claude and Gemini to evaluate our datasets. The LLM was not involved in data generation, model training, or writing of technical content.

Table 5: Examples of the four prompt strategies used for causal discovery tasks in VLMs evaluation.

| PROMPT STRATEGY | TEMPLATE EXAMPLE |
|---|---|
| BASIC | **ANALYZE THE PROVIDED IMAGES AND IDENTIFY CAUSAL RELATION-SHIPS BETWEEN THE VARIABLES.** COMPLETE THE CAUSALITY ADJACENCY MATRIX BASED ON THE IDENTIFIED RELATIONSHIPS AND BRIEFLY EXPLAIN YOUR CONCLUSIONS. THERE ARE {VARIABLES}: X, Y, Z. PLEASE FILL THIS CAUSALITY ADJACENCY MATRIX: $$\begin{bmatrix} - & - & - \\ - & - & - \\ - & - & - \end{bmatrix}$$ IN THIS MATRIX, `MATRIX[I][J] = 1` MEANS VARIABLE $i$ CAUSES VARIABLE $j$, WHILE `MATRIX[I][J] = 0` MEANS THERE IS NO DIRECT CAUSAL RELATIONSHIP. |
| EXPLICIT FUNCTION | **YOU ARE A CAUSAL DISCOVERY EXPERT. YOUR OBJECTIVE IS TO AN-ALYZE THE PROVIDED IMAGES AND IDENTIFY ANY CAUSAL RELATION-SHIPS BETWEEN THE VARIABLES.** USE THE IDENTIFIED RELATIONSHIPS TO COMPLETE THE CAUSALITY ADJACENCY MATRIX AND PROVIDE A BRIEF EXPLANATION SUPPORTING YOUR CONCLUSIONS. THERE ARE VARIABLES: X, Y, Z ... |
| ZERO-SHOT-COT | ANALYZE THE PROVIDED IMAGES AND IDENTIFY CAUSAL RELATION-SHIPS BETWEEN THE VARIABLES ... **LET'S THINK STEP BY STEP** ... |
| FEW-SHOT | ANALYZE THE PROVIDED IMAGES AND IDENTIFY CAUSAL RELATION-SHIPS BETWEEN THE VARIABLES ... **EXAMPLE 1**: TO DETERMINE THE CAUSAL RELATIONSHIPS BETWEEN THE SPRING CONSTANT, WEIGHT, AND DEFORMATION OF THE SPRING, WE CAN USE HOOKE'S LAW, WHICH STATES THAT THE FORCE EXERTED BY A SPRING IS DIRECTLY PROPORTIONAL TO THE DEFORMATION (DIS-PLACEMENT) OF THE SPRING, GIVEN BY: $$F = k \cdot x$$ WHERE: <ul><li>$F$ IS THE FORCE APPLIED (RELATED TO WEIGHT),</li><li>$k$ IS THE SPRING CONSTANT,</li><li>$x$ IS THE DEFORMATION OF THE SPRING.</li></ul> FROM THIS, WE CAN INFER: 1. SPRING CONSTANT $k$ AFFECTS THE DE-FORMATION OF THE SPRING ($x$): IF THE SPRING CONSTANT INCREASES, FOR THE SAME WEIGHT, THE DEFORMATION DECREASES. 2. WEIGHT AFFECTS **DEFORMATION OF THE SPRING** ($x$): AN INCREASE IN WEIGHT CAUSES MORE DEFORMATION. 3. THE SPRING CONSTANT ($k$) AND **WEIGHT** DO NOT DIRECTLY AFFECT EACH OTHER. BASED ON THESE RELATIONSHIPS, THE CAUSALITY ADJACENCY MATRIX IS: $$\begin{bmatrix} 0 & 0 & 1 \\ 0 & 0 & 1 \\ 0 & 0 & 0 \end{bmatrix}$$ **EXPLANATION:** <ul><li>ELEMENT (1,3) IS 1 BECAUSE THE SPRING CONSTANT AFFECTS DEFORMATION.</li><li>ELEMENT (2,3) IS 1 BECAUSE THE WEIGHT AFFECTS DEFORMA-TION.</li><li>THE OTHER ENTRIES ARE 0 BECAUSE THERE IS NO DIRECT CAUSAL RELATIONSHIP OTHERWISE.</li></ul> **EXAMPLE 2:** ...; **EXAMPLE 3:** ... ; |

