# OpenReview forum: "Causal3D: a comprehensive benchmark for causal learning from visual data"
_ICLR.cc/2026/Conference — Submitted to ICLR 2026_

### Official Review · Reviewer_gU3A · 2025-10-23

**Soundness:** 2
**Presentation:** 2
**Contribution:** 3
**Rating:** 4
**Confidence:** 3

**Summary:**

Paper introduces Causal3D, a new benchmark that serves to evaluate causal reasoning from observational visual data. Benchmark includes 19 realistic 3D scene datasets comprising of physically consistent "real-world" scenes and hypothetical scenes.
Work aims to bridge traditional causal discovery and computer vision by aligning tabular data with visual representations. The dataset combines three main causal tasks, namely Causal discovery from tabular data, Causal representation learning from images and Causal discovery & intervention from few images.
Authors also provide comprehensive evaluation across traditional causal discovery algorithms, causal representation learning models, and modern Vision-Language Models (VLMs) such as ChatGPT, Gemini, and Claude across the various tasks.

**Strengths:**

CAUSAL3D is the first dataset combining explicit causal graphs and realistic 3D visual representations, filling a key gap in causal learning evaluation.

Work has a clear motivation in aiming to bridge the gap, with comprehensive evaluations for each task across several methods.
The authors benchmark a diverse range of models, from classical causal discovery to VAE-based causal representation learning and commercial VLMs.

Inclusion of multi-view images and analysis on the impact gives a novel perspective into causal discovery from from visual representations.

**Weaknesses:**

1. Despite claims of physical consistency, the benchmark remains entirely simulation-based. Simply using realistic backgrounds does not equate to real-world data capture, and this should be stated more cautiously.
2. Claims of different fundamental physical principles (line 208 -211) with differing number of variables and unique causal structures. Would like to have seen more statistics and details on the dataset to support the claims in "different dimensions, including the number of variables (ranging from 2 to 5), multiple causal structures, different (linear/nonlinear) types of causal relations, and various camera views and backgrounds in 3D scenes"
3. Under section 4.3 of Causal Representation Learning discussion is only on 2 qualitative results that are presented for evaluation. It would have been more beneficial if analysis were based on more samples across the entire dataset.
4. Again under Views and Backgrounds, the analysis would have been better supported with qualitative results. i.e multi-view vs single-across different scenes, and performance virtual and realistic backgrounds.

Minor comments\
Generally in section 4, it would have been clearer to have represented the numerical values of the results tabular.

**Questions:**

My main concern lies with the claims and analysis of the performance of current methods on the benchmark to support claims of the benefits of the dataset.

The work as a whole is well motivated and could serve as a foundational benchmark for causal reasoning research within computer vision. However, the current empirical analysis does not sufficiently substantiate the paper’s claims.
The causal representation learning section is particularly weak, relying almost entirely on qualitative samples without quantitative validation or comprehensive evaluation across the dataset. Furthermore, the lack of comprehensive details and analysis on their own dataset makes it difficult to substantiate their claims as stated in point 2 of weakness.

---

> ### Author Response · Authors · 2025-11-21
>
> We appreciate the reviewers' feedback and will give the following responses correspondingly.
> ## 1.Existence of Real–Synthetic Gap
> We appreciate the reviewer’s concern and would like to clarify the steps we have taken to mitigate it. Although Causal3D is simulated to provide full access to the underlying causal model (an essential property for benchmarking and rarely attainable with real-world data), we have made deliberate efforts to minimize discrepancies from real environments. Our simulations on physical scenes are **physically consistent, follow real-world physical laws, and are rendered with high visual fidelity in Blender**. This ensures that object appearances, lighting, and interactions resemble real conditions, allowing models to learn from causal cues that are relevant and potentially transferable.
>
> Besides, we emphasize that the use of synthetic data does not diminish the contribution of Causal3D at the current stage of research. **First**, Causal3D provides a novel benchmark bridging 3D computer vision and causal inference, something largely missing in prior work. **Second**, synthetic data is essential for ensuring full controllability over both causal graphs and visual factors. Notably, our experiments show that state-of-the-art methods struggle in this controlled setting; let alone real-world scenarios with uncontrolled variability. We will include this clarification in future versions.
> ## 2.Missing of Dataset statistics
> In the submitted version, we provide **detailed descriptions of each scene in Appendix 6** (L212). Because each scene is quite specific, we felt that coarse, high-level statistics would be of limited value, and instead chose to give detailed information of each setting, explicitly highlighting the number of variables and the causal structures.
>
> Here, for a global overview of Causal3D, we include a summary table below reporting the **number of settings** under the different number of variables (and related statistics) in each scene.
> Number of Nodes|2|3|4|5
> -|-|-|-|-
> **Realistic scene**|2|2|2|2
> **Hypothetical scene**|2|3|3|3
>
> A summary table reporting the **number of causal structures** under the different number of variables (and related statistics) in each scene:
> Number of Nodes|2|3|4|5
> -|-|-|-|-
> **Realistic scene**|1|2|2|2
> **Hypothetical scene**|1|2|2|2
>
> We will add such tables to the Appendix in the revised version.
> ## 3.Only 2 Qualitative Results in Representation Learning
> We have conducted evaluations on the additional scenes, and the results are **consistent** with those presented in Sec.4.3—and in some cases, performance is even lower, reflecting the current limitations of visual causal learning methods. We selected the scenes shown in the paper since they are **representative** and offer clear visualizations that effectively illustrate our main findings. We will consider including a broader range of results in the appendix to further support the generality of our claims.
> ## 4.Missing of Qualitative Results in Sec.4
> We appreciate the reviewer’s observation. In the original paper, we selected representative figures to clearly illustrate the comparison between the single- and multi-view settings. Based on our observation on other settings, **our findings remain consistent** across different configurations, including both virtual and realistic backgrounds. In the revised version, we will include additional results in the Appendix to further strengthen the claim.
> ## 5.The Main Concern about Soundness of the Claims
> From our understanding, the main concern derives from 1. insufficient quantitative experimental results; 2. missing of detailed dataset statistics.
> **Quantitative results**:
> - In the submitted version, we have provided quantitative results in Fig. 3, Fig. 5, and Tab. 2 for comprehensive causal learning assessment in different settings (covering various causal methods and all of our datasets).
> - We now further include quantitative measures for causal representation learning to better assess how well the learned latent representations align with the ground truth. We introduce **Maximum Information Coefficient(MIC)**. This metric indicates the degree of information relevance between the learned representation and the ground truth labels of concepts. The closer MIC is to 1, the higher the degree of relevance. Conversely, 0 indicates low the relevance.
> MIC|Reflection|Spring|Seesaw|Magnet
> -|-|-|-|-
> CausalVAE|0.99|0.82|0.68|0.92
> DEAR|0.20|0.18|0.24|0.23
> ICM-VAE|0.99|0.48|0.69|0.83
>
> We found that the latent representations learned by most models generally align well with the ground‑truth labels, though there remains significant room for improvement.
>
> **Dataset details**:
> Appendix 6 has included a detailed description of every dataset in Causal3D. In addition, we will provide more statistics for Causal3D and add a concise overview (summary tables like our reponse to Weakness 2) to more clearly and quickly convey the organization of the benchmark.

---

### Official Review · Reviewer_oeR1 · 2025-10-29

**Soundness:** 3
**Presentation:** 3
**Contribution:** 3
**Rating:** 4
**Confidence:** 4

**Summary:**

This paper introduces Causal3D, a comprehensive benchmark designed to evaluate causal reasoning capabilities from visual data. The benchmark is built on 19 distinct 3D-rendered scenes that combine realistic imagery with explicit, underlying causal structures (DAGs and structural equations). The authors conduct extensive evaluations across three causal tasks—causal discovery from tabular data, causal representation learning from images, and causal discovery from few images using VLMs—demonstrating the benchmark's utility and revealing significant challenges for current methods, especially as causal complexity increases.

**Strengths:**

1)	The introduction of Causal3D fills a significant void. Existing datasets are either purely tabular, lack explicit causal graphs, or are too simplistic. Causal3D's integration of realistic 3D visuals with rigorous causal ground truth is a major contribution that will facilitate research at the intersection of computer vision and causal inference.

2)	The benchmark is meticulously designed for progressive evaluation. The inclusion of multiple views and backgrounds (real/virtual) allows for nuanced analysis of model robustness to spurious correlations, a critical aspect of causal learning.

**Weaknesses:**

1)   The benchmark, in its current form, is focused solely on observational data. A crucial aspect of causal inference is interventional and counterfactual reasoning. The authors acknowledge this in the limitations, but its absence currently restricts the benchmark's scope. Incorporating interventional data (e.g., by allowing users to manipulate variables in the 3D engine) would make it even more powerful.

2)	The evaluation of VLMs on the causal discovery task, while interesting, could be clarified. The prompt asks the model to output an adjacency matrix, which is a highly structured and non-standard output for these models. It would be beneficial to discuss potential parsing issues and how the models' raw text outputs were consistently mapped to a matrix format. Furthermore, a comparison against a simple baseline (e.g., a model that always predicts no edges) would help contextualize the absolute performance scores, which are often quite low.

3)	The paper would benefit from a more explicit justification for why the specific 8 real-world and 11 hypothetical scenes were chosen. A discussion on how they cover a representative space of causal structures (e.g., chains, forks, colliders) would strengthen the claim of comprehensiveness.

4） For the causal representation learning section, the choice of evaluation scenes for each model (e.g., why CDG-VAE was evaluated on reflection and pendulum but not spring) is mentioned but could be more clearly motivated upfront.


5） Figures 3, 5, and 6 are referenced in the text but were not included in the provided content. Their absence makes it difficult to fully assess the experimental findings. The review assumes these figures effectively support the claims.

6）The analysis of multi-view vs. single-view performance is insightful but could be taken a step further. A hypothesis or deeper discussion on why multi-view helps in complex scenes but hurts in simple ones would be valuable (e.g., is it due to overfitting, or the introduction of conflicting visual cues?).

**Questions:**

See the weaknesses.

---

> ### Author Response · Authors · 2025-11-21
>
> We sincerely appreciate the valuable comment of reviewer oeR1, and we will further justify and clarify our work.
>
> ## 1.Missing interventional and counterfactual reasoning
> We would like to clarify that although the current version of Causal3D does not explicitly include interventional or counterfactual data, they can be readily **generated due to our access to the full underlying causal model**. As demonstrated in Section 4.5, our benchmark already supports causal intervention testing, and extending it to provide explicit interventional and counterfactual data is straightforward.
>
> Importantly, Causal3D is a novel benchmark that combines explicit causal structure with realistic 3D visual observations, a setting that closely mirrors real-world complexity. Our results show that even state-of-the-art methods struggle significantly in this observational data, highlighting the value and challenge our benchmark presents, even without requiring access to interventional data.
>
>
> ## 2. Parsing issues of VLMs' output
> General VLMs like ChatGPT have the capacity to return the answer following the given template. As shown in Tab.3 in our appendix, we explicitly ask VLMs to fill the blank adjacency matrix of the causal graph, and parser return matrix (given template) for reliable quantification.
> From our best understanding, the reviewer may misunderstand how we parse the VLMs’ output. During the evaluation, we do not parse the textual description of causal relation, but parse the return, fix matrix format. Which is more stable and reliable of quantification.
>
> ## 3. The Choice of Scenes
> We appreciate the suggestion. Our goal in selecting the 8 real-world and 11 hypothetical scenes was not to exhaustively enumerate all possible graphs, **but to cover a diverse set of canonical causal motifs** that are standard in the causality literature, such as chains (A $\rightarrow$ B $\rightarrow$ C), forks / common causes (A $\rightarrow$ B, A $\rightarrow$ C), colliders (A $\rightarrow$ C $\leftarrow$ B), as well as combinations involving mediators, confounders, and independent causes.
> The real-world scenes were chosen to ground these motifs in visually realistic, physically plausible situations (e.g., multi-object interactions), while the hypothetical scenes allow us to systematically instantiate cleaner graph topologies without additional visual confounds. In the revision, we will add this short statement to clarify how this collection jointly spans a representative space of basic causal patterns.
>
> ## 4. The choice of scene in representation learning.
> In causal representation learning, different methods impose different requirements on the input format, so the choice of scene depends on the **compatibility between each method and the scene**. Specifically, CDG-VAE requires masking half of the raw input image during training, which is easy to realize in the Reflection scene (where a symmetric layout allows a clean half-image mask). In contrast, the Spring and Seesaw scenes are incompatible: the cause and effect cannot be cleanly separated by a simple half-image mask.
>
> ## 5. Missing reference of Fig.3,5,6 in the main paper
> We are a bit puzzled by this comment. In the submitted paper, we do refer to Fig. 3 at line 302, Fig. 5 at line 386, and Fig. 6 at line 408. There may be some misunderstanding, and we would be grateful if the reviewer could clarify what is missing or unclear.
>
> ## 6. More discussion about  multi-view vs. single-view
> We appreciate the reviewer’s observation and agree that the comparison between single-view and multi-view settings reveals important insights into the behavior of current VLMs in causal discovery.
>
> In the multi-view setting, the same underlying causal mechanism is presented through visually varied realizations. In simple scenes, this variation can introduce **redundant or conflicting** cues, which act as noise for VLMs when identifying a consistent causal law (e.g., Hooke’s law), especially when such laws may already be memorized by the model. Instead of reasoning over diverse observations, the model may rely on **superficial visual features** to retrieve **pre-learned** causal associations, leading to reduced performance.
>
> In contrast, in complex scenes, multi-view input is often beneficial, as it helps expose hidden variables, resolve occlusions, and disambiguate causal structure. Our results show that VLMs do benefit from multi-view input in these cases, but still struggle to consistently integrate information across views in a causally meaningful way.
>
> For humans, aggregating multiple views generally improves causal understanding. The fact that VLMs exhibit the opposite trend in simple cases and only partial gains in complex ones highlights a fundamental limitation: **current VLMs lack robust mechanisms for view-consistent causal reasoning, and often depend on pattern matching rather than structural inference.**

---

### Official Review · Reviewer_BbKk · 2025-10-31

**Soundness:** 2
**Presentation:** 2
**Contribution:** 2
**Rating:** 4
**Confidence:** 3

**Summary:**

This paper introduces Causal3D, a new benchmark for causal learning from visual data. It includes 19 realistic and hypothetical 3D scenes combining images, tabular data, and causal graphs. The benchmark aims to test models on causal discovery, representation learning, and intervention. The dataset is well-structured and systematically designed. Experiments with classical methods, causal VAEs, and recent methods highlight current limitations in visual causal reasoning.

**Strengths:**

- The dataset is comprehensive covering both physical and hypothetical causal structures.
- Several types of experiments with various methods (tabular, image, multimodal).
- The work aim to provide insides on the limitations of current causal and vision-language models.

**Weaknesses:**

- The benchmark still deals with simple, static scenes with few objects and low-dimensional latent variables. It lacks dynamic data or temporal causality, which limits real-world relevance.
- There is no reference or comparison to “Weakly Supervised Causal Representation Learning” or to the Causal Circuit Dataset, which are relevant benchmarks.
- The evaluation scope focuses mainly on 2D rendered scenes and does not test robustness under more realistic 3D dynamics or noise.
- It is unclear how to extend the benchmark to higher-dimensional or real-world videos.

**Questions:**

- Do you plan to extend Causal3D to dynamic or temporal datasets?
- Could you discuss why prior benchmarks like Causal Circuit (Weakly Supervised Causal Representation Learning) were not referenced or compared?

---

> ### Author Response · Authors · 2025-11-21
>
> We sincerely appreciate the reviewer’s constructive comments and recognition.
>
> ## 1. The benchmark still deals with simple, static scenes with few objects and low-dimensional latent variables. It lacks dynamic data or temporal causality, which limits real-world relevance.
>
> Our work aims to create a foundational benchmark for evaluating models’ capabilities in visual causal learning. To the best of our knowledge, this is the first 3D causal dataset of this scale with explicit ground-truth causal models. Constructing such a dataset is far from simple:
> (1) **Selecting appropriate scenes was nontrivial**, as all causal variables must be visually observable in rendered images to support evaluation. This excluded many abstract or non-visual causal factors (e.g., sound, force, electricity).
> (2) **Creating physically consistent, high-resolution 3D scenes** required significant rendering effort and computational resources, especially given the need for precise control over causal variables and interventional settings.
>
> We agree that incorporating temporal dynamics and more variables is a valuable next step. Nonetheless, we believe the current version serves as an important stepping stone, providing a controlled, scalable platform for benchmarking key aspects of causal reasoning in visual data, which are important for real-world applications. We believe it can pave the way for more complex benchmarks and inspire further research in this area.
>
> ## 2. There is no reference or comparison to “Weakly Supervised Causal Representation Learning” and causal circuit dataset.
>
> The paper _“Weakly Supervised Causal Representation Learning”_ adopts a weakly supervised setting, where the data does not use labels for variables, but instead relies on image pairs captured before and after unknown interventions. **This setup significantly differs** from other methods evaluated in our paper using labeled causal variables, as well as the main focus of our benchmark. Because the supervision regimes and assumptions are incompatible, comparing these methods directly would be unfair and uninformative.
>
> Regarding the causal circuit dataset, our benchmark differs in several key aspects:
> (1) **Diversity and structure:** our dataset includes more diverse and well-structured design, including varied backgrounds, controlled changes in the number of causal variables, and multiple types of causal relationships, etc.
> (2) **3D scene construction:** our work focuses on causal learning in 3D environments. To our knowledge, the first work in this area, offering a unique testbed for visual causal learning.
>
> ## 3. The evaluation scope focuses mainly on 2D rendered scenes and does not test robustness under more realistic 3D dynamics or noise.
> Our benchmark provides **multi-view 2D renderings**, a common and practical representation of 3D scenes, and we have corresponding multi-view evaluation in Fig. 6. Most evaluations focus on 2D scenes because most existing causal learning models (e.g., CausalVAE) are originally designed for single 2D images, and even vision-language models typically process videos as sequences of 2D frames. While recent 3D models (e.g., point cloud autoencoders) support geometric understanding, they lack causal reasoning capabilities and are not yet compatible with intervention-based evaluations. Therefore, our 2D evaluation aligns with the current capabilities of causal learning models.
>
> Our simulation includes **diverse, realistic** 3D backgrounds (with realistic noises) and **physically meaningful** laws, making it a valuable resource for **robust testing** of causal learning models in settings that closely approximate **real-world** conditions.
>
> ## 4. It is unclear how to extend the benchmark to higher-dimensional or real-world videos.
>
> We must clarify that extending our data to real-world videos is not difficult. Although our data is simulated, the objects involved and the underlying physical principles are grounded in reality. Therefore, by **preparing the corresponding physical components in a laboratory setting and demonstrating the relevant physical phenomena,** it would be feasible to record videos, even using handheld devices.
>
> As for high-dimensional data, we can increase the dimensionality of exogenous variables by **introducing more irrelevant objects or by making the background more complex.** For endogenous variables, higher dimensionality can be achieved by incorporating **temporal information, introducing dynamic objects (e.g. dynamically demonstrating the compression of a spring or the projectile motion of a ball.),** or **constructing more complex physical scenes.** We plan to progressively expand in this direction in future versions of the dataset.

---

### Official Review · Reviewer_n8uL · 2025-11-03

**Soundness:** 3
**Presentation:** 3
**Contribution:** 3
**Rating:** 6
**Confidence:** 3

**Summary:**

This paper presents CAUSAL3D, a novel benchmark designed to evaluate causal learning from complex visual data. CAUSAL3D includes multiple physically consistent and hypothetical scenes, providing data in both tabular and image modalities. Furthermore, it supports intervention settings. The authors conduct a comprehensive evaluation, benchmarking traditional causal discovery algorithms and LLM-based methods. The evaluation involves several scenarios including Causal Discovery from Tabular Data, Causal Representation Learning, and Causal Discovery from Few Images, providing critical insights into the limitations of current SOTA models.

**Strengths:**

- The dataset covers a diverse range of scenarios, including both tabular and image-based data, physically consistent scenes and hypothetical scenes.
- The evaluation is comprehensive, benchmarking both traditional causal discovery methods and modern LLM-based approaches across multiple tasks.
- The paper is well-written and clear.

**Weaknesses:**

- The paper fails to discuss the  Synthesis and Reality gap. It is unclear whether this synthetic dataset can accurately reflect causal relationships found in real-world data or if it can benefit research on real-world problems.
- The discussion on the construction of the causal data lacks detail regarding the noise term. It is not specified how the noise term is defined or how different noise levels might impact the evaluation results.

**Questions:**

- Could the authors discuss the dataset's generalization to real-world cases? For example, can models trained on this dataset using supervised learning successfully perform causal inference on real data?
- Could the authors provide a more detailed discussion of the noise term? For instance, it would be helpful to see evaluation results under different noise strengths.

---

> ### Author Response · Authors · 2025-11-21
>
> We thank the reviewer for the thoughtful feedback and valuable suggestions.
> _(The weakness and question raised by this reviewer are aligned, so we only present the question here.)_
>
>
> ## 1. Generalization to real-world cases. For example, can models trained on this dataset using supervised learning successfully perform causal inference on real data?
> Although the dataset is simulated to provide access to the full causal model, which is crucial for benchmarking but rarely available in real-world data, we have taken careful steps to **minimize the gap between simulation and reality**. The simulations are **physically consistent, grounded in real-world physical laws** and set in **visually realistic environments**. We use Blender to render scenes with high visual fidelity, ensuring that object appearances and interactions closely resemble real-world conditions. This enables models to learn from causal cues that are relevant and transferable to real settings.
> While complete out-of-distribution deployment to real data may require adaptation, our dataset offers a valuable testbed for developing and pretraining models that can later be adapted to real data via fine-tuning, domain adaptation, or causal transfer learning methods. Thus, our benchmark provides an important step toward **scalable and generalizable** causal learning in **realistic** environments.
>
>
> ## 2. Definition of noise term and evaluation under different noise strengths
> **Purpose and Definition of Noise.** We introduce additive noise primarily to ensure **identifiability** of causal models. Prior work has established that (i) **linear models with non‑Gaussian additive noise can uniquely identify causal direction** [1], and (ii) this result extends to **nonlinear models with any additive noise** [2]. Following these theoretical foundations, we adopt uniform noise, a simple and effective non‑Gaussian choice, to guarantee identifiable causal mechanisms while keeping the simulation stable.
>
> **Different Noise Strengths.** While generating entirely new datasets at multiple noise levels would require several days of rendering, theory and prior empirical work indicate that both **extremely large and extremely small noise variances can hinder causal direction recovery**. Within our benchmark, the current noise configuration is chosen to balance identifiability and visual/physical realism. Importantly, the primary contribution of our work is the construction of a systematical causal 3D benchmark and its validation using representative baseline models—not a systematic study of noise sensitivity. The clear performance trends observed in our experiments (e.g., decreasing accuracy as the causal graph grows in complexity) indicate that our dataset design and **noise settings are sound** and aligned with causal theory.
>
> [1] Shimizu, Shohei, et al. "A linear non-Gaussian acyclic model for causal discovery." Journal of Machine Learning Research 7.10 (2006).
> [2] Hoyer, Patrik, et al. "Nonlinear causal discovery with additive noise models." Advances in neural information processing systems 21 (2008).

---

### Meta-Review · Area_Chair_AuKw · 2026-01-18

**Summary:**

The paper introduces Causal3D, a benchmark for evaluating causal learning from visual data by pairing rendered 3D scenes with explicit causal graphs. Reviewers agreed that the problem is well motivated and that the benchmark design is systematic, covering multiple causal learning settings and model classes. However, significant concerns were raised about the realism, scope, and empirical substantiation of the benchmark. In particular, reviewers questioned whether the static, synthetic scenes meaningfully reflect real world causal complexity, whether the evaluation, especially for causal representation learning, is sufficiently comprehensive and quantitative, and whether the paper’s claims about the benchmark’s breadth and impact are fully supported by the presented analysis.

**Reviewer Concerns:**

Several reviewer concerns were clarified in the rebuttal, but in the AC’s assessment, they are not fully resolved in the current submission. The authors explained the motivation for using simulated data, emphasizing the need for full access to ground truth causal models and arguing that the scenes are physically consistent and visually realistic. They also clarified the role of additive noise using established causal identifiability theory and explained why a single noise configuration was chosen. Questions about scene selection and coverage of canonical causal motifs were addressed conceptually, and additional dataset statistics and quantitative metrics such as MIC for causal representation learning were provided.

However, based on the reviews and the rebuttal, substantial limitations remain in the scope and evidence presented. The benchmark is restricted to simple, static scenes with a small number of variables and does not include explicit temporal, dynamic, or counterfactual data. While the authors discussed future extensions, these aspects are not supported in the current version. In addition, although additional quantitative analysis was added, the evaluation of causal representation learning remains limited in breadth, with incomplete coverage across scenes and continued reliance on selected examples. The gap between synthetic visual data and real world causal reasoning is discussed conceptually but is not empirically evaluated.

As a result, while the rebuttal improves clarity and justification, the AC judges that the core concerns raised in the reviews are not fully addressed by the current submission.

**Reviewer Scores:**

Reviewers were not given an opportunity to participate in post-rebuttal discussion. The following reflects the AC’s judgment of how scores might have changed based on the rebuttal and revisions.

Reviewer n8uL (original score: 6): Likely unchanged. While the rebuttal clarified the use of synthetic data and the noise model, concerns about real world generalization were addressed conceptually rather than empirically.

Reviewer BbKk (original score: 4): Likely unchanged. The core concerns regarding static scenes, lack of temporal dynamics, and limited realism reflect fundamental scope limitations of the current benchmark.

Reviewer oeR1 (original score: 4): Possibly a slight increase but still below threshold. The rebuttal provided detailed clarifications on interventions, scene selection, and evaluation design, but these primarily justify design choices rather than resolving structural limitations.

Reviewer gU3A (original score: 4): Likely unchanged. Although additional quantitative results and dataset statistics were provided, concerns about the strength and completeness of the empirical analysis remain for the current submission.

Overall, even accounting for the rebuttal, the AC does not expect a shift toward a clear acceptance consensus.

---

### Decision · Program_Chairs · 2026-01-26

Reject